Corrected: Author correction

# Smac mimetics and oncolytic viruses synergize in driving anticancer T-cell responses through complementary mechanisms

Dae-Sun Kim[1,2,3], Himika Dastidar[1,2,3], Chunfen Zhang[1,2], Franz J. Zemp[1,2,3], Keith Lau[1,2,3,4], Matthias Ernst[1,2], Andrea Rakic[1,2,5], Saif Sikdar[1,2,3], Jahanara Rajwani[1,2], Victor Naumenko[1,2,3,4], Dale R. Balce[6], Ben W. Ewanchuk[7], Pankaj Tailor[6], Robin M. Yates[6,7], Craig Jenne[3,4], Chris Gafuik[1,2,3] & Douglas J. Mahoney[1,2,3,7]

Second mitochondrial activator of caspase (Smac)-mimetic compounds and oncolytic viruses were developed to kill cancer cells directly. However, Smac-mimetic compound and oncolytic virus therapies also modulate host immune responses in ways we hypothesized would complement one another in promoting anticancer T-cell immunity. We show that Smac-mimetic compound and oncolytic virus therapies synergize in driving CD8[+] T-cell responses toward tumors through distinct activities. Smac-mimetic compound treatment with LCL161 reinvigorates exhausted CD8[+] T cells within immunosuppressed tumors by targeting tumor-associated macrophages for M1-like polarization. Oncolytic virus treatment with vesicular stomatitis virus (VSV$^{\Delta M51}$) promotes CD8[+] T-cell accumulation within tumors and CD8[+] T-cell activation within the tumor-draining lymph node. When combined, LCL161 and VSV$^{\Delta M51}$ therapy engenders CD8[+] T-cell-mediated tumor control in several aggressive mouse models of cancer. Smac-mimetic compound and oncolytic virus therapies are both in clinical development and their combination therapy represents a promising approach for promoting anticancer T-cell immunity.

[1] Alberta Children's Hospital Research Institute, Calgary, AB, Canada T2N 4N1. [2] Arnie Charbonneau Cancer Institute, Calgary, AB, Canada T2N 4N1. [3] Department of Microbiology, Immunology and Infectious Disease, Faculty of Medicine, University of Calgary, Calgary, AB, Canada T2N 4N1. [4] Snyder Institute for Chronic Disease, Calgary, AB, Canada T2N 4N1. [5] Department of Medical Sciences, Faculty of Medicine, University of Calgary, Calgary, AB, Canada T2N 4N1. [6] Department of Comparative Biology and Experimental Medicine, Faculty of Veterinary Medicine, University of Calgary, Calgary, AB, Canada T2N 4N1. [7] Department of Biochemistry and Molecular Biology, Faculty of Medicine, University of Calgary, Calgary, AB, Canada T2N 4N1. Correspondence and requests for materials should be addressed to D.J.M. (email: djmahone@ucalgary.ca)

Therapies targeting a patient's adaptive immune system have been validated for treating cancer and represent one of the most significant advances in clinical oncology in decades[1]. While monotherapies are highly efficacious in a small percentage of patients, rationally designed combination therapies have shown activity in a higher proportion of clinical trial participants[2,3]. These exciting results provide a strong justification for treating cancer with multiple therapies that engender antitumor T-cell activity in distinct yet complementary ways.

Smac-mimetic compounds (SMCs) and oncolytic viruses (OVs) were recently shown to synergize in promoting tumor regression in mouse models of cancer[4]. SMCs comprise a group of small molecules designed to antagonize the inhibitor of apoptosis (IAP) proteins and sensitize cancer cells to death triggered by inflammatory cytokines such as tumor necrosis factor alpha (TNFα)[5]. OVs represent a group of natural and engineered viruses developed to selectively infect and kill tumors based on genetic defects inherent to cancer cells[6]. Cell culture studies suggested that the anticancer synergy between SMC and OV therapies is due to apoptosis of SMC-treated cancer cells, triggered by TNFα secreted during the OV infection[4]. However, both SMC and OV therapies are potent immunostimulants[7–10]. This prompted us to investigate whether their combined treatment may function in vivo by promoting anticancer immunity.

Here we show that SMC and OV therapies synergize in treating immunogenic tumors by driving anticancer T-cell responses through complementary mechanisms. Studies in mouse models demonstrate that SMC therapy indirectly rejuvenates exhausted CD8+ T cells by targeting tumor-associated macrophages (TAM) for M1-like polarization, while OV therapy promotes CD8+ T-cell recruitment and serves as a non-specific immune system adjuvant. Surprisingly, we found that TNFα-mediated cancer cell killing through its canonical receptor TNFR1 is not required for anticancer immunity and therapeutic response in vivo. Finally, SMC/OV therapy is further enhanced by immune checkpoint blockade (ICB), using αPD-1 antibodies, with triple SMC/OV/ICB therapy leading to long-term tumor regression in nearly 90% of tumor-bearing mice.

## Results

**T-cell dependence of LCL161 and VSV$^{\Delta M51}$ combination therapy.** As both SMC and OV therapies have been shown to promote T-cell activity[7–10], we hypothesized that their combined treatment in vivo may function by promoting a more robust anticancer immune response. To test this, we first asked whether outcomes to SMC (LCL161)[11] and OV (vesicular stomatitis virus, VSV$^{\Delta M51}$)[12] combination therapy (ref. [4] and Supplementary Figs. 1 and 22) are dependent upon T-cell activity. T-cell neutralizing antibodies were administered to immunocompetent Balb/c mice bearing orthotopic EMT6 breast carcinoma prior to LCL161 ± VSV$^{\Delta M51}$ treatment. CD8+ cell depletion completely abrogated the therapeutic effect of LCL161 ± VSV$^{\Delta M51}$ (Fig. 1a and Supplementary Fig. 2). Intriguingly, CD4+ cell depletion induced profound anticancer activity on its own (Fig. 1b and Supplementary Fig. 3). These results demonstrate that LCL161 and VSV$^{\Delta M51}$ co-therapy induces EMT6 tumor regression by engaging CD8+ T-cell-dependent anticancer immunity.

As the synergy between SMCs and OVs was previously attributed to TNFα-triggered apoptosis of cancer cells[4], we sought to determine whether TNFα-mediated cancer cell death stimulates the curative anticancer immunity generated by the combination therapy. We therefore knocked out TNFR1 from EMT6 cells using clustered regularly interspaced short palindromic repeats/CRISPR-associated protein-9 nuclease (CRISPR/Cas9) and tested for responsiveness to LCL161 + VSV$^{\Delta M51}$. While EMT6$^{TNFR1−/−}$ cells (clones 2–10 and 3–12) grown in culture were completely resistant to LCL161 + TNFα induced cell death, as expected (Fig. 1c and Supplementary Figs. 4 and 22), they maintained complete responsiveness to the combination therapy when grown as tumors in vivo (Fig. 1d–f). Indeed, when a small panel of mouse cancer cells was evaluated for sensitivity to LCL161 + TNFα and three additional lines chosen for in vivo testing based on degree of sensitivity (Supplementary Figs. 5 and 22) and known immunogenicity[13–15] (Supplementary Fig. 6), the shared characteristic among the responsive (EMT6 breast carcinoma and M3-9-M rhabdomyosarcoma) vs. resistant (76–9 rhabdomyosarcoma and 4T1 breast carcinoma) tumors in vivo was immunogenicity, not sensitivity to LCL161 + TNFα in vitro (Fig. 1g–i). Moreover, mice bearing M3-9-M tumors lost all responsiveness to LCL161 + VSV$^{\Delta M51}$ treatment when depleted of either CD8+ or CD4+ T cells (Fig. 1i). Collectively, these results revealed that LCL161 + VSV$^{\Delta M51}$ induces T-cell-mediated regression of tumors completely unresponsive to LCL161 + TNFα killing in vitro, which in our hands (Supplementary Figs. 5, 7, and 23) and others[16,17] represents the majority of cancer cell lines tested.

The divergent role of CD4+ T cells in the M3-9-M vs. EMT6 models prompted us to explore their phenotype within these tumors. Flow cytometry showed that the majority of CD4+ tumor-infiltrating lymphocytes (TILs) in M3-9-M tumors have an activated phenotype (CD4+CD25+Foxp3−; Supplementary Fig. 8). In contrast, most CD4+ TILs in EMT6 tumors are naive (CD4+CD25−Foxp3−; Supplementary Fig. 8), a phenotype recently shown to be a precursor of regulatory T cells in breast cancer[18]. While it is not clear why these differences exist between M3-9-M and EMT6 tumors, they offer a potential explanation for the opposing effect of CD4+ T-cell depletion on LCL161 + VSV$^{\Delta M51}$-induced tumor regression in these two model systems (Fig. 1b, i).

**LCL161 reinvigorates exhausted T cells within the TME.** While LCL161 treatment alone caused some degree of T-cell-dependent tumor regression, VSV$^{\Delta M51}$ monotherapy had only a minimal effect (Supplementary Fig. 1). With respect to the combination therapy, this suggested to us that LCL161 might be the primary driver of the antitumor immunity, while VSV$^{\Delta M51}$ might serve to enhance or prime for a better response. To ascertain how LCL161 might elicit anticancer immunity, we first studied its effect on T cells within the tumor microenvironment (TME) and tumor-draining lymph node (TdLN). Because T cells are commonly exhausted within the TME[19], we asked whether LCL161 acts to reverse TIL exhaustion. CD8+ T cells isolated from established EMT6 tumors showed both functional and molecular indications of exhaustion, as measured by unresponsiveness to phorbol myristate acetate (PMA)/ionomycin stimulation ex vivo (Fig. 2a and Supplementary Fig. 9) and membrane expression of programmed cell death protein 1 (PD-1) and T-cell immunoglobulin and mucin-domain containing-3 (Tim-3) (Fig. 2b and Supplementary Fig. 10), respectively[20]. However, 20–40% of CD8+ T cells isolated from LCL161-treated tumors stained positive for interferon (IFN)-γ or TNFα after PMA/ionomycin stimulation (Fig. 2a), suggesting a partial reversal of TIL exhaustion. In contrast, LCL161 treatment had no bearing on the number of CD8+ T cells infiltrating EMT6 tumors (Fig. 2c). Within the TdLN, CD8+ T cells were less exhausted, as measured functionally and by surface marker expression (Fig. 2d, e), and LCL161 treatment affected neither their antigen- or PMA/ionomycin-mediated activation, nor their overall cell number (Fig. 2d, f–g and Supplementary Fig. 11). To determine whether LCL161 was acting directly on TILs to reverse their exhausted state, we purified CD8+ T cells from EMT6 tumors and measured PMA/ionomycin-induced cytokine expression ±

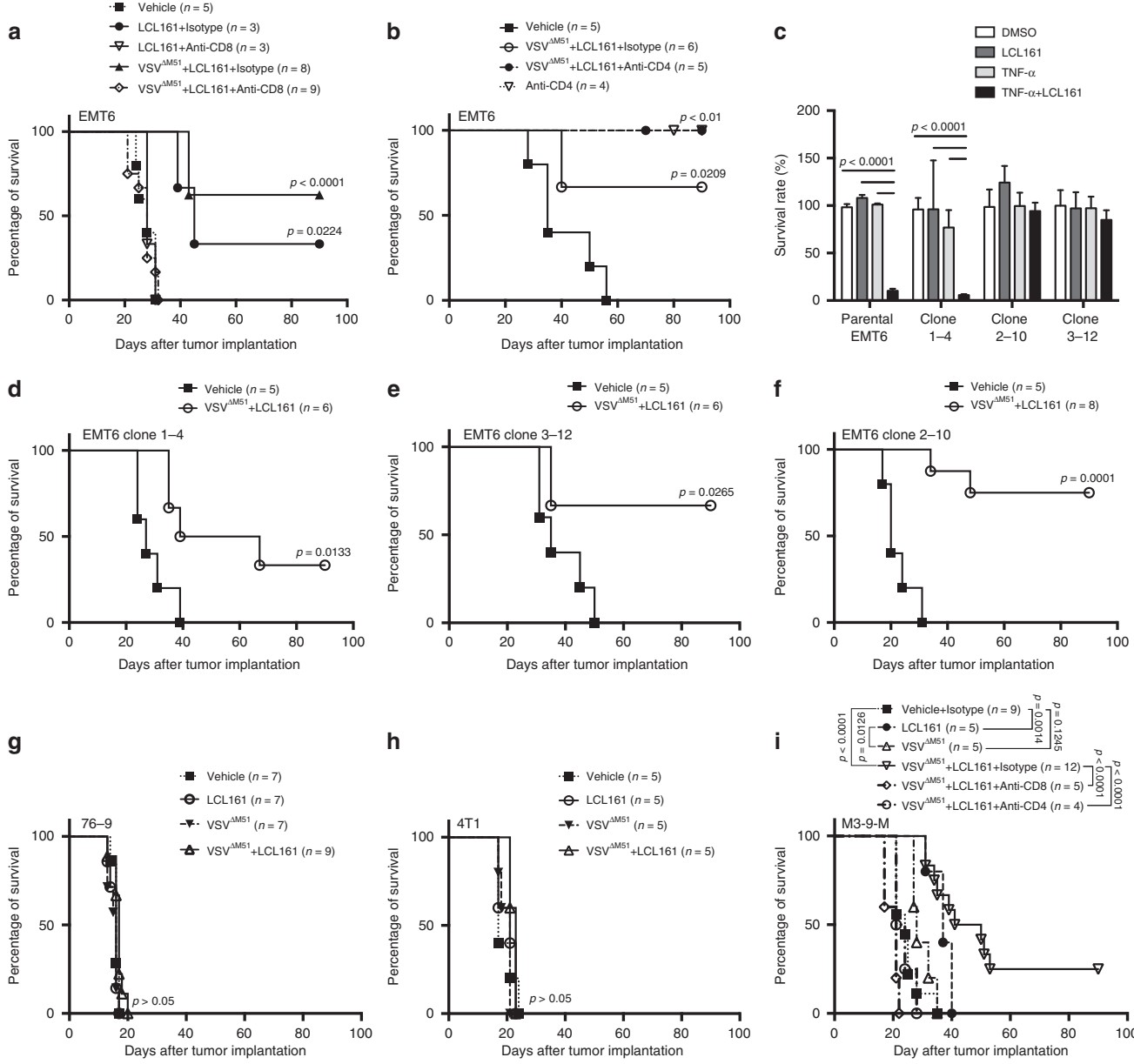

**Fig. 1** LCL161 and VSV$^{\Delta M51}$ combination therapy induces CD8$^+$ T-cell-mediated tumor regression independent of TNFR1 signaling in cancer cells. **a** Overall survival of EMT6 tumor-bearing mice treated with LCL161 ± VSV$^{\Delta M51}$ ± CD8 neutralizing antibody (or isotype control; triplicate experiments; log-rank test). **b** Overall survival of EMT6 tumor-bearing mice treated with LCL161 + VSV$^{\Delta M51}$ ± CD4 neutralizing antibody (or isotype control; duplicate experiments; log-rank test). **c** Cell viability of parental EMT6 cells and three EMT6$^{TNFR1-CRISPR}$ clones assayed for TNFR1 bioactivity by treatment with LCL161 + TNFα (100 ng mL$^{-1}$), measured by Alamar Blue 48 h later ($n = 3$ biological replicates per experiment; triplicate experiments; mean ± SD; ANOVA with Tukey's multiple comparisons test). **d–f** Overall survival of EMT6$^{TNFR1+/+}$ (**d** clone 1-4) and EMT6$^{TNFR1-/-}$ (**e**, **f** clones 2-10 and 3-12) bearing mice treated with LCL161 + VSV$^{\Delta M51}$ (duplicate experiments; log-rank test). **g–i** Overall survival of 76-9 **g**, 4T1 **h** and M3-9-M **i** tumor-bearing mice treated with LCL161 + VSV$^{\Delta M51}$ (M3-9-M: triplicate experiments; 76-9 and 4T1: single experiment). Effect of CD4 or CD8 (or isotype control) neutralization is shown for M3-9-M (single experiment; log-rank test)

LCL161 treatment. Treating TILs directly with LCL161 had no bearing on their functionally exhausted state (Fig. 2h). We conclude from these data that LCL161 partially reinvigorates exhausted CD8$^+$ T cells within the immunosuppressed TME through a CD8$^+$ T-cell non-autonomous mechanism.

**LCL161 polarizes TAM toward M1-like.** Because TIL exhaustion is commonly the result of immunosuppressive signals emanating from tumor-infiltrating immune cells[19], we asked whether LCL161 therapy promotes TIL reinvigoration in vivo by creating an immunosupportive cytokine milieu within EMT6 tumors.

Indeed, numerous proinflammatory and T-helper type 1 (Th1) chemokines (e.g., RANTES, MCP-1, and MIP-2; Fig. 3a) and cytokines (e.g., IL-6, IFNγ, and IL-1; Fig. 3b) accumulated within the TME after LCL161 treatment, whereas several immunosuppressive cytokines (IL-10 and IL-4; Fig. 3c) were reduced. Similar results were obtained in EMT6$^{TNFR1-/-}$ tumors (Supplementary Fig. 12). Moreover, a type I IFN gene signature associated with anticancer immunity[21] was found within EMT6 tumors after LCL161 treatment (Fig. 3d). These data demonstrate that LCL161 can alter the cytokine profile within tumors toward one more supportive of anticancer immunity.

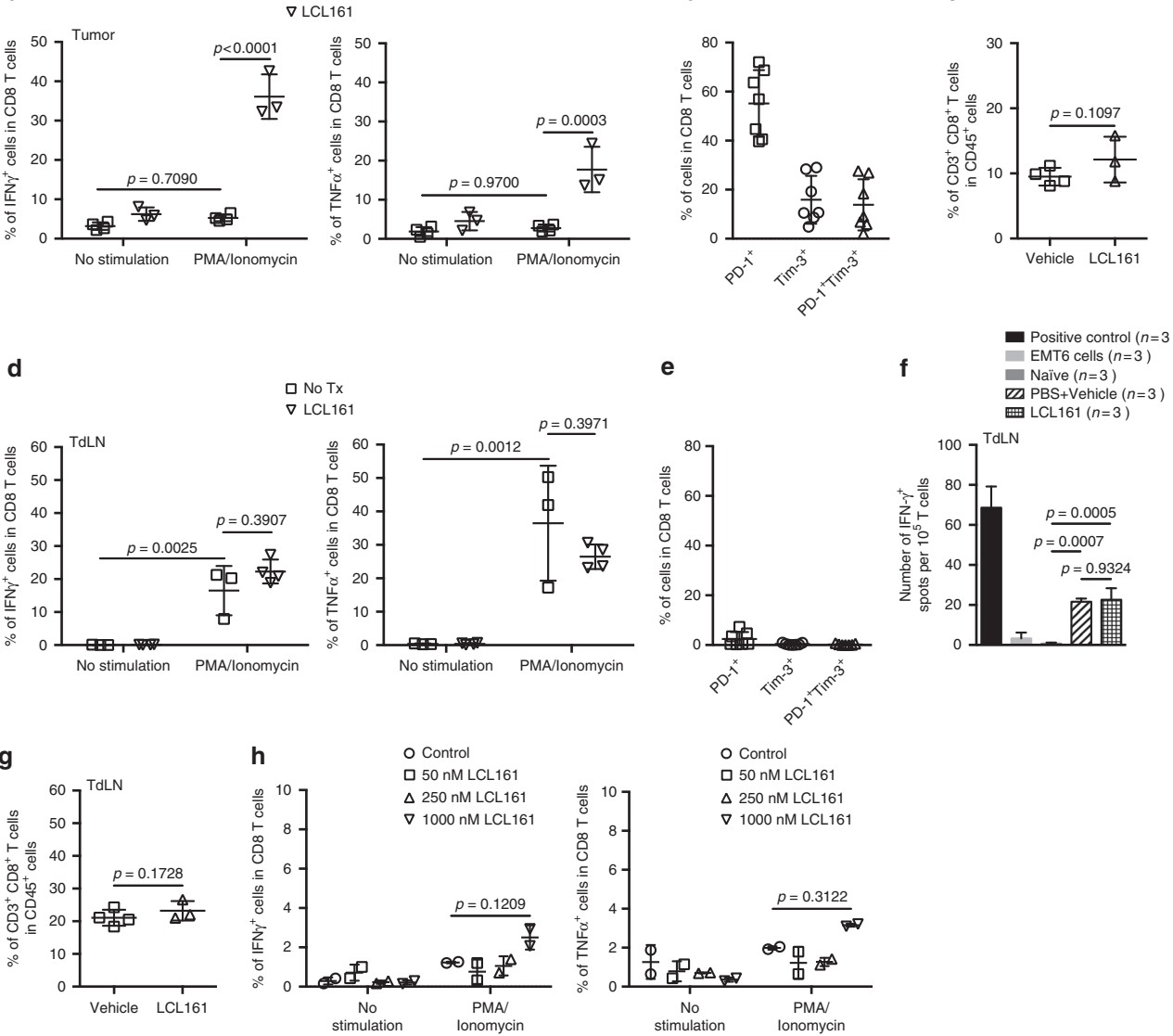

**Fig. 2** LCL161 therapy rescues CD8[+] T-cell exhaustion within the TME. **a** Intracellular staining for IFNγ or TNFα within CD8[+] T cells isolated from EMT6 tumors 7 days post treatment and stimulated with PMA and ionomycin ex vivo, measured by flow cytometry (duplicate experiments; mean ± SD; ANOVA with Bonferroni's multiple comparisons test). **b** Membrane staining for PD-1 and Tim-3 expression on CD8[+] T cells isolated from 12-day-old EMT6 tumors, measured by flow cytometry (duplicate experiments; mean ± SD). **c** CD8[+] T cells within EMT6 tumors 7 days post treatment, measured by flow cytometry (triplicate experiments; mean ± SD; t-test). **d** Intracellular staining for IFNγ or TNFα within CD8[+] T cells isolated from EMT6 TdLN 7 days post treatment and stimulated with PMA and ionomycin, measured by flow cytometry (single experiment; mean ± SD; ANOVA with Bonferroni's multiple comparisons test). **e** Membrane staining for PD-1 and Tim-3 expression on CD8[+] T cells isolated from 12-day-old EMT6 TdLNs, measured by flow cytometry (duplicate experiment; mean ± SD). **f** Activation of CD8[+] T cells isolated from TdLN 8 days post treatment, after co-culture with EMT6 cells, measured by IFNγ ELISpot assay (n = 3 biological replicates, single experiment; mean ± SD; ANOVA with Tukey's multiple comparisons test). **g** CD8[+] T cells within EMT6 TdLN 7 days post treatment, measured by flow cytometry (duplicate experiments; mean ± SD; t-test). **h** Intracellular staining for IFNγ or TNFα within CD8[+] T cells isolated from 12-day-old EMT6 tumors and treated in vitro with LCL161 at the indicated concentrations for 24 h, measured by flow cytometry (n = 2 biological replicates, single experiment; mean ± SD; ANOVA with Bonferroni's multiple comparisons test)

Interestingly, many of the LCL161-altered cytokines are known to be produced by, or regulate the activation state of, TAMs[22–24]. TAMs are commonly polarized within the TME toward an alternative or M2-like phenotype, which can suppress T-cell activity and promote T-cell exhaustion[19]. Because: (1) EMT6 tumors are highly enriched with TAMs (ref. [25] and Supplementary Fig. 13), (2) IAP antagonism is known to promote macrophage cell death[26], and (3) IAP antagonism was recently shown to promote TAM activation/M1-like polarization in mouse models of multiple myeloma[27] and ovarian carcinoma[28], we hypothesized that TAM depletion and/or

M1-like polarization may underlie the genesis of an immuno-supportive TME by LCL161 treatment. To address these possibilities, we first asked whether LCL161 treatment caused a loss of TAMs in vivo. As hypothesized, LCL161 led to the partial depletion of CD45[+]F4/80[+] TAMs and some subsets of CD45[+]CD11b[+]MHC-II[±]Ly6C[±] monocytes/macrophages[29] within established EMT6 tumors in vivo (although its ability to directly kill macrophages in vitro was minimal and only observed at high concentrations; Supplementary Figs. 14 and 15). We therefore tested whether TAM depletion by intratumor clodro-nate liposome (CL) treatment could phenocopy the anticancer

effect of LCL161. While CL depleted TAMs within EMT tumors to a similar extent as LCL161 (Supplementary Fig. 16a, b), it had minimal anticancer effect on its own, did not synergize with $VSV^{\Delta M51}$ treatment, and attenuated LCL161-mediated tumor regression (Supplementary Fig. 16c). This result suggests that the partial depletion of TAMs by LCL161 does not play a significant role in its anticancer function. In contrast, it appears that at least some TAMs are required for EMT6 tumor regression after LCL161 treatment, consistent with a previous report studying peptide vaccines[30].

We therefore asked whether LCL161 affected TAM polarization. Indeed, LCL161 treatment led a shift in TAM orientation within EMT6 tumors away from M2-like, as measured by a reduction in the immunosuppressive subsets of $CD45^{+}CD11b^{+}$ MHC-II$^{\pm}$Ly6C$^{\pm}$ cells expressing arginase-1 (Fig. 3e)[29, 31]. To determine whether LCL161 could mediate macrophage polarization directly, we treated cultured bone marrow-derived macrophages (BMDMs) with LCL161 and measured a series of functional and molecular markers of activation/polarization. These experiments showed that LCL161 treatment: (1) increases

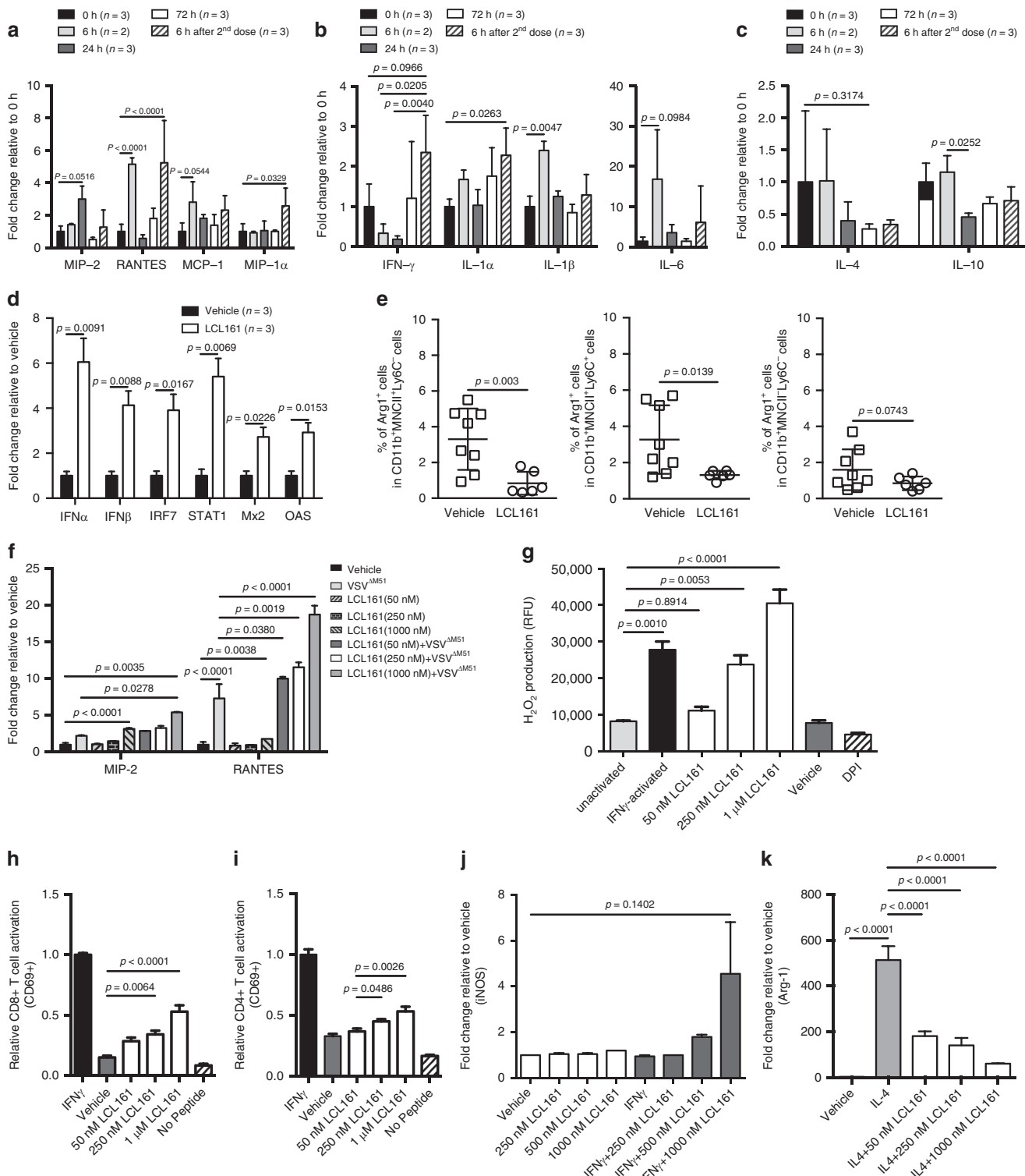

the cell surface expression of class I and II MHC, CD40L, and CD80 (Supplementary Fig. 17); (2) leads to the secretion of proinflammatory cytokines, and their enhanced secretion upon VSV-inoculation (Fig. 3f); (3) increases reactive oxygen species (ROS) generation during phagocytosis (Fig. 3g); (4) heightens activation of T cells in class I and II MHC-restricted antigen presentation assays (Fig. 3h, i); (5) promotes inducible nitric oxide synthase (iNOS) expression upon treatment with the M1-polarizing agent IFNγ (Fig. 3j); and (6) blunts arginase-1 expression after treatment with the M2-polarizing agent IL-4 (Fig. 3k). Collectively, these measures demonstrate that LCL161 elicits cell-autonomous alterations in macrophage orientation toward an activated or M1-like state, and away from an immunosuppressive or M2-like state[22–24].

**VSV promotes TIL accumulation and is an immune adjuvant.** We next sought to understand the contribution of VSV$^{ΔM51}$ to the combination therapy. We hypothesized that the virus could serve multiple roles, ranging from a beacon for T-cell recruitment to infected tumors, to a modulator of immunosuppression with the TME, to an adjuvant for T-cell activation within the TdLN. To begin these studies, we probed the cytokine response to VSV$^{ΔM51}$ treatment within EMT6 tumors using multiplex ELISA and found that the T-cell activating cytokine IFNα and the canonical T-cell recruiting chemokines IP-10 (CXCL10) and MIG (CXCL9) were enriched by 6 h post treatment (Fig. 4a). While EMT6 tumors are relatively resistant to OV infection in vivo[25, 32], a small number of infectious virions were detected 12 h post-i.v. delivery (Supplementary Fig. 18). Cell culture experiments showed that EMT6 cells, macrophages, and cancer-associated fibroblasts all secrete inflammatory cytokines in response to VSV$^{ΔM51}$ treatment in vitro (Supplementary Fig. 19), and we speculate they may each contribute to the cytokine response generated within EMT6 tumors in vivo. Importantly, an accumulation of CD8$^+$ T cells was found within VSV$^{ΔM51}$-treated EMT6 tumors, and this effect was strongest when the virus was combined with LCL161 (Fig. 4b). Interestingly, VSV$^{ΔM51}$ had no bearing on the activation of tumor-infiltrating CD8$^+$ T cells by PMA and ionomycin (Fig. 4c) nor the orientation of TAM (Fig. 4d). When combined with LCL161, the virus elicited a similar increase in the number of IFNγ- or TNFα-producing T cells within EMT6 tumors as compared to LCL161 alone (Figs. 2a and 4c). Similar data were obtained in EMT6$^{TNFR1−/−}$ tumors (Supplementary Fig. 20). We conclude from these results that a VSV$^{ΔM51}$ infection of EMT6 tumors induces cytokine/chemokine secretion that promotes CD8$^+$ T-cell infiltration within EMT6 tumors without altering TAM polarization or TIL exhaustion, a mechanism distinct and complementary to that invoked by LCL161.

We next asked whether the virus could act as a non-specific adjuvant for anticancer T-cell activation within the TdLN. Indeed, VSV$^{ΔM51}$ treatment led to a significant increase in EMT6-reactive CD8$^+$ T cells in the TdLN when combined with LCL161 (Fig. 4e), without altering their overall numbers (Supplementary Fig. 21). The virus was found by intravital microscopy (IVM) to robustly infect cells within the TdLN that resemble subscapular sinus (SCS) macrophages and dendritic cells (DCs), with some of the infection overlapping with the SCS surface marker CD169 (Fig. 4f, *left panel*). A qualitatively similar pattern of infection was observed in the spleen (Fig. 4f, *right panel*). CD169$^+$ macrophages and DCs in secondary lymphoid organs are known to produce large quantities of cytokines upon virus encounter[33], and predictably VSV$^{ΔM51}$ led to high levels of immunostimulatory cytokines secreted into blood (Fig. 4g). Taken together, these measures are consistent with a role for VSV$^{ΔM51}$ as a non-specific immune system adjuvant for anticancer T-cell responses by infecting antigen-presenting cells (APCs) within secondary lymphoid organs and promoting immunostimulatory cytokine production.

**SMC synergizes with VSV and αPD-1 therapy.** Intriguingly, we found the timing of VSV$^{ΔM51}$ treatment relative to LCL161 to be critical for therapeutic success, with experiments in which LCL161 treatment preceded VSV$^{ΔM51}$ demonstrating no therapeutic synergy between the two agents (Fig. 5a). As SMC treatment was shown previously to attenuate antiviral cytokine production from cells grown in culture[34], we speculated that LCL161 pretreatment might decrease immunomodulatory cytokine secretion by VSV$^{ΔM51}$. Indeed, the immunostimulatory cytokine response to VSV$^{ΔM51}$ therapy in vivo was dampened by LCL161 pretreatment (Fig. 5b). However, a direct effect of LCL161 on VSV$^{ΔM51}$-induced antiviral signaling in vitro was not observed (Fig. 5c), consistent with recent data from Beug and colleagues[4]. We therefore asked whether the VSV$^{ΔM51}$ infection itself was altered by LCL161 treatment, as a potential explanation for the attenuated cytokine response. LCL161 pretreatment led to a complete loss of VSV$^{ΔM51}$ infection within the tumor and TdLN, and a tenfold reduction in viral productivity in the spleen, as measured by plaque assay (Fig. 5d). While the mechanisms underlying this observation are not known, it may be related to the cytokines produced by LCL161 treatment (Fig. 3a–d), many of which have potent antiviral activity against rhabdoviruses[12, 35]. Regardless, these results highlight the importance of timing the SMC and OV treatments strategically, and more broadly the need for in-depth understanding of the specific mechanisms involved in combination immunotherapies as a prelude to designing their dosing schedule.

---

**Fig. 3** LCL161 creates and immunosupportive TME by polarizing TAM toward M1-like. **a–c** Expression levels of immune-promoting chemokines **a** and cytokines **b** and immunosuppressive cytokines **c** within the interstitial fluid of 12-day-old EMT6 tumors, measured by Luminex (duplicate experiments; mean ± SD; ANOVA with Tukey's multiple comparisons test). **d** mRNA expression of type I interferon (IFN) and IFN-stimulated genes in EMT6 tumors 7 days after LCL161 treatment initiation, measured by qPCR (single experiment; mean ± SD; ANOVA with Tukey's multiple comparisons test). **e** Intracellular staining for arginase-1 in CD11b$^+$MHC-II$^±$Ly6C$^±$ TAMs isolated from EMT6 tumors 72 h post treatment (duplicate experiments; mean ± SD; t-test). **f** Immune-promoting cytokines measured in BMDM cell culture media 12 h after LCL161 treatment ± VSV$^{ΔM51}$ treatment (MOI 10, added at 6 h), measured by Luminex (n = 3 biological replicates, single experiment; mean ± SD; ANOVA with Tukey's multiple comparisons test). **g** Extracellular H$_2$O$_2$ production in BMDMs following phagocytosis of serum-opsonized zymosan particles, measured by oxidation of the Amplex UltraRed reagent (n = 3 biological replicates per experiment, duplicate experiments; mean ± SD; ANOVA with Tukey's multiple comparisons test). **h** CD69 surface expression on transgenic OT-1 T cells co-cultured with BMDM pulsed with full-length OVA, measured by flow cytometry (n = 3 biological replicates per experiment, duplicate experiments; mean ± SD; ANOVA with Tukey's multiple comparisons test). **i** CD69 surface expression on transgenic 2D2 T cells co-cultured with BMDM pulsed with MOG$_{35-55}$ peptide, measured by flow cytometry (n = 3 biological replicates per experiment, duplicate experiments; mean ± SD; ANOVA with Tukey's multiple comparisons test). **j** iNOS expression in BMDMs treated for 40 h with the M1-polarizing agent IFNγ ± LCL161 at the indicated concentrations, measured by flow cytometry (n = 3 biological replicates per experiment, duplicate experiment; mean ± SD; ANOVA with Tukey's multiple comparisons test). **k** Arginase-1 expression in BMDMs treated for 40 h with the M2-polarizing agent IL-4 ± LCL161 at the indicated concentrations, measured by qPCR (n = 3 biological replicates, single experiment; mean ± SD; ANOVA with Tukey's multiple comparisons test)

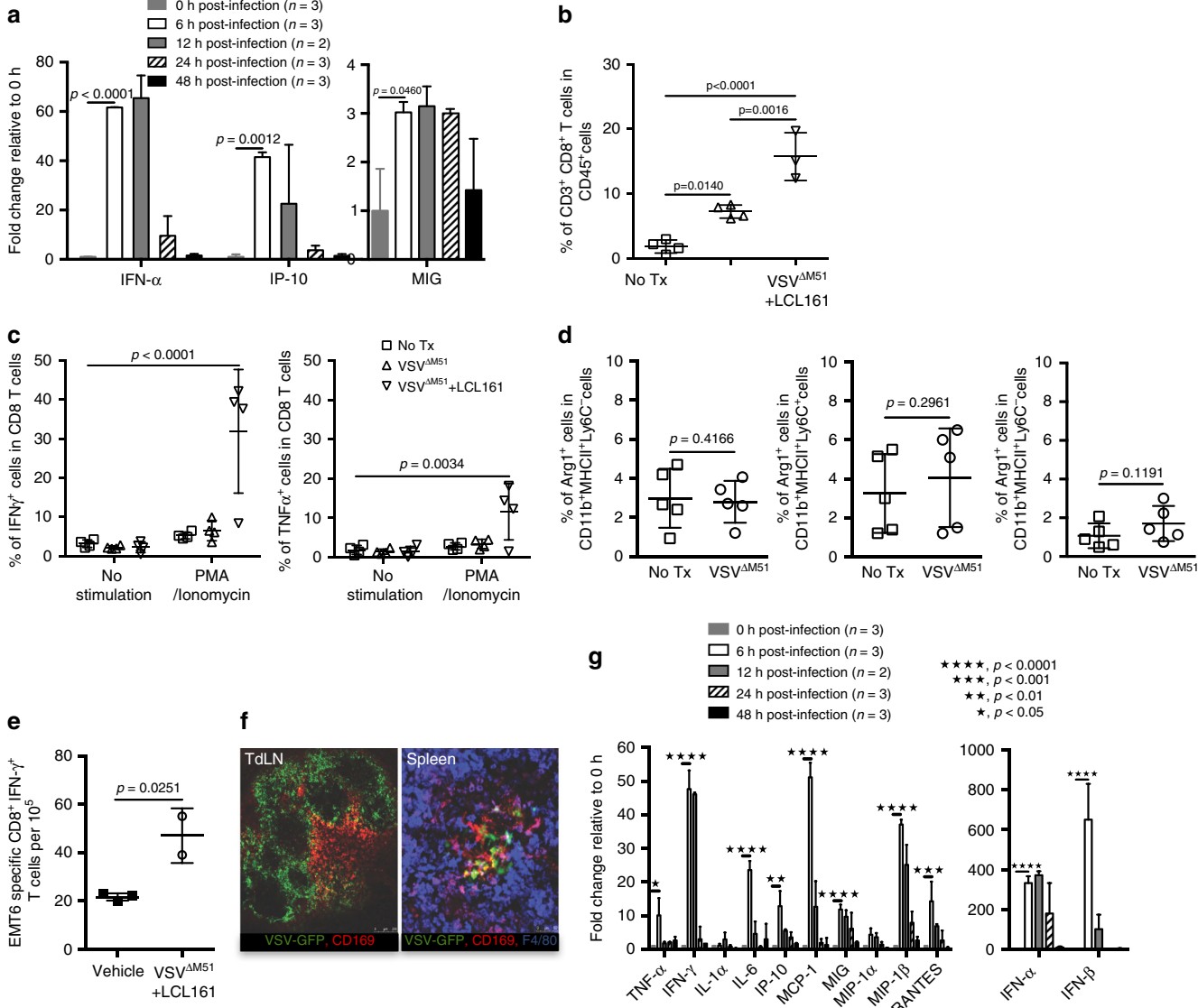

**Fig. 4** VSV$^{\Delta M51}$ promotes T-cell accumulation within tumors and serves as a systemic immune system adjuvant. **a** Expression levels of T-cell promoting chemokines and cytokines within the interstitial fluid of EMT6 tumors, measured by Luminex (duplicate experiments; mean ± SD; ANOVA with Tukey's multiple comparisons test). **b** CD8$^+$ T cells within EMT6 tumors 7 days post treatment, measured by flow cytometry (triplicate experiments; mean ± SD; ANOVA with Tukey's multiple comparisons test). **c** Intracellular staining for IFNγ or TNFα within CD8$^+$ T cells isolated from EMT6 tumors 7 days post treatment and stimulated with PMA and ionomycin ex vivo, measured by flow cytometry (duplicate experiment; mean ± SD; ANOVA with Tukey's multiple comparisons test). **d** Intracellular staining for arginase-1 in CD11b$^+$MHC-II$^\pm$Ly6C$^\pm$ TAMs isolated from EMT6 tumors 72 h post treatment (single experiment; mean ± SD; t-test). **e** Activation of CD8$^+$ T cells isolated from EMT6 tumors 8 days post treatment, after co-culture with EMT6 cells, measured by IFNγ ELISpot assay (single experiment; mean ± SD; t-test). **f** Representative IVM images of VSV$^{\Delta M51\text{-GFP}}$ infection of TdLN and spleen, taken 8 h post-VSV$^{\Delta M51\text{-}}$GFP treatment (n = 3 biological replicates). Scale bars for TdLN and spleen = 250 μM and 50 μM, respectively. **g** Cytokines within blood taken from EMT6 tumor-bearing mice treated with VSV$^{\Delta M51}$, measured by Luminex (single experiment; mean ± SD; ANOVA with Tukey's multiple comparisons test)

**Synergy between LCL161, VSV and αPD-1 therapy.** Finally, because LCL161 did not completely reverse T-cell exhaustion (Fig. 2a), we asked whether outcomes to LCL161 + VSV$^{\Delta M51}$ could be further enhanced by addition of αPD-1 therapy. Indeed, nearly 90% of EMT6-bearing mice treated with triple LCL161 + VSV$^{\Delta M51}$ + αPD-1 therapy showed a complete and durable tumor response (Fig. 6a). As shown in Fig. 2b, CD8$^+$ TIL in EMT6 tumors have high expression of PD-1. We also found that EMT6 cells grown in culture constitutively and IFNγ inducibly express PD-L1 (Fig. 6b). Moreover, PD-L1 was highly expressed on TAM subsets within EMT6 tumors in vivo, and could be detected on a smaller percentage of the CD45$^-$ fraction within EMT6 tumors (i.e., tumor and stromal cells) (Fig. 6c). Consistent with our functional studies of CD8$^+$ TIL

exhaustion (Fig. 2a), the combination therapy induced a partial decrease in PD-1 expression on CD8$^+$ TIL (Fig. 6d). In contrast, PD-1 levels increased in CD4$^+$ T cells within both tumor and TdLN (Fig. 6d, e). LCL161 treatment led to decreased PD-L1 expression in some but not all subsets of TAMs (Fig. 6f). Taken together, these data show that PD-1 inhibition improves outcomes to LCL161 + VSV$^{\Delta M51}$, which we speculate is because the PD-1/PD-L1 signaling axis remains partially intact within the TME after LCL161 + VSV$^{\Delta M51}$ treatment.

## Discussion

SMCs have been in clinical development for over a decade. Unfortunately, they have shown minimal efficacy in patient

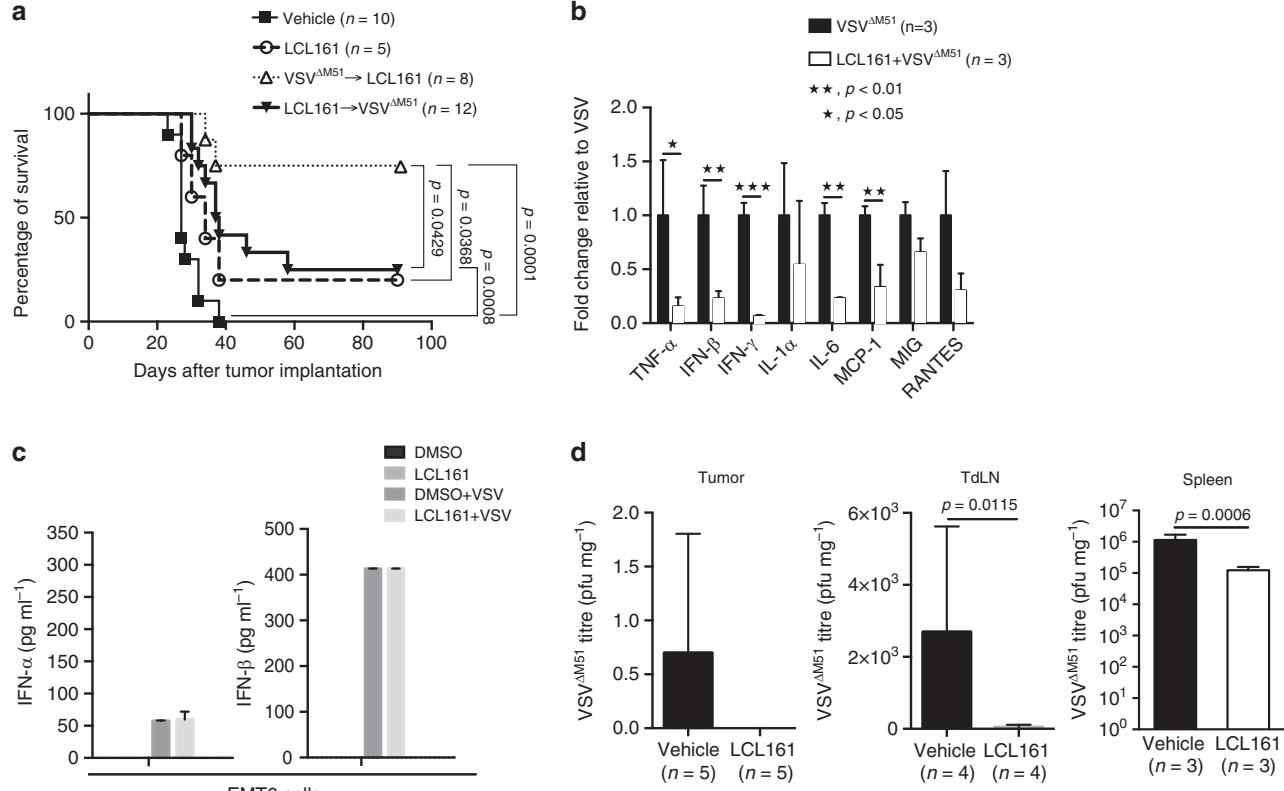

**Fig. 5** Anticancer synergy between LCL161 and VSV$^{\Delta M51}$ is dose schedule dependent. **a** Overall survival of EMT6 tumor-bearing mice treated with LCL161 ± VSV$^{\Delta M51}$ at different dosing schedules (duplicate experiments; log-rank test). **b** Cytokines within blood taken from EMT6 tumor-bearing mice treated with VSV$^{\Delta M51}$ for 24 h, after 72 h pretreatment with LCL161 (or vehicle), measured by Luminex (single experiment; mean ± SD; $t$-test). **c** Concentration of type I IFN within EMT6 cell culture media 24 h after VSV$^{\Delta M51}$±LCL161 pretreatment for 2 h, measured by ELISA ($n = 3$ biological replicates per experiment; duplicate experiments; mean ± SD). **d** Infectious VSV$^{\Delta M51}$ particles isolated from tumor, TdLN, and spleen of EMT6 tumor-bearing mice 12 h after VSV$^{\Delta M51}$ treatment ± LCL161 pretreatment for 72 h, measured by plaque assay (spleen, duplicate experiments; tumor and TdLN, single experiment; mean ± SD; $t$-test)

trials[36]. SMC therapy has long been viewed through the lens of sensitizing cancer cells to death triggers—mostly TNFα—a mechanism defined and studied almost exclusively in cell culture experiments[4, 37–39]. A key and surprising finding from our study is that TNFα-TNFR1 signaling in the cancer cell is not required for LCL161 + VSV$^{\Delta M51}$-mediated tumor regression in vivo, in stark contrast to its critical importance in vitro[4]. This suggests that a cancer cell-centric view of SMC therapy does not capture its major anticancer mechanisms in vivo, at least not in mouse models of cancer. Rather, our data and that of other recent studies in mouse models of multiple myeloma and glioma point toward the genesis of an immunosupportive TME and the enhancement of anticancer T-cell responses as the key mechanisms underlying tumor regression following SMC therapy[27, 40]. While our study identifies a role for SMCs in acting directly on TAM to polarize toward M1-like and relieve TIL exhaustion, a recent study by Chesi et al.[27] showed that SMCs can promote TAM activation indirectly, via secretion of immunomodulatory factors from cancer cells. This was shown to promote cancer cell phagocytosis by the activated macrophage and the eventual generation of anticancer T-cell immunity. Indeed, earlier studies showed that SMC therapy can also promote co-stimulation of adoptively transferred T cells[8] and enhance DC activation in tumors[7]. Our study, therefore, adds insight into a growing understanding of SMC therapies as potent, multi-mechanistic modulators of anticancer T-cell responses within the TME.

There is much excitement around combination immunotherapy as a strategy to shift the balance within the cancer-immunity

relationship toward the immune system[41]. Consistent with this idea, we show that SMC immunotherapy synergizes with OV (±ICB) immunotherapy in causing tumor regression. Recent studies have also shown therapeutic synergy between SMC and ICB, adoptive T-cell and toll-like receptor (TLR) agonist immunotherapies[8, 27, 40]. Our data support a mechanism, whereby OV infection of the tumor and secondary lymphoid organs promotes T-cell recruitment and activation, respectively, which synergizes with the immunomodulatory effects of SMCs on TAM orientation. Importantly, we report that the order in which each agent is administered is critical, with synergy being observed only if the OV therapy precedes the SMC therapy. We propose this occurs because of SMC-induced secretion of immune cytokines, some of which can establish an antiviral state prior to OV treatment. This would serve to attenuate the OV infection within the tumor and secondary lymphoid organs, and the immunostimulatory sequelae that follow. While the importance of dosing schedule on outcomes to other SMC combination immunotherapies is not yet known, our results suggest that it should be carefully considered, especially given the recent registration of two SMC-ICB combination therapy clinical trials (NCT02890069 and NCT02587962).

Broadly speaking, an increasing understanding of SMCs as an immunomodulatory agent and its multimodal mechanisms of action within the TME, coupled with observations of profound synergy with other cancer immunotherapies, such as OV and ICB, is creating a renewed excitement for SMCs in the treatment of cancer. We propose that refocusing the study, development

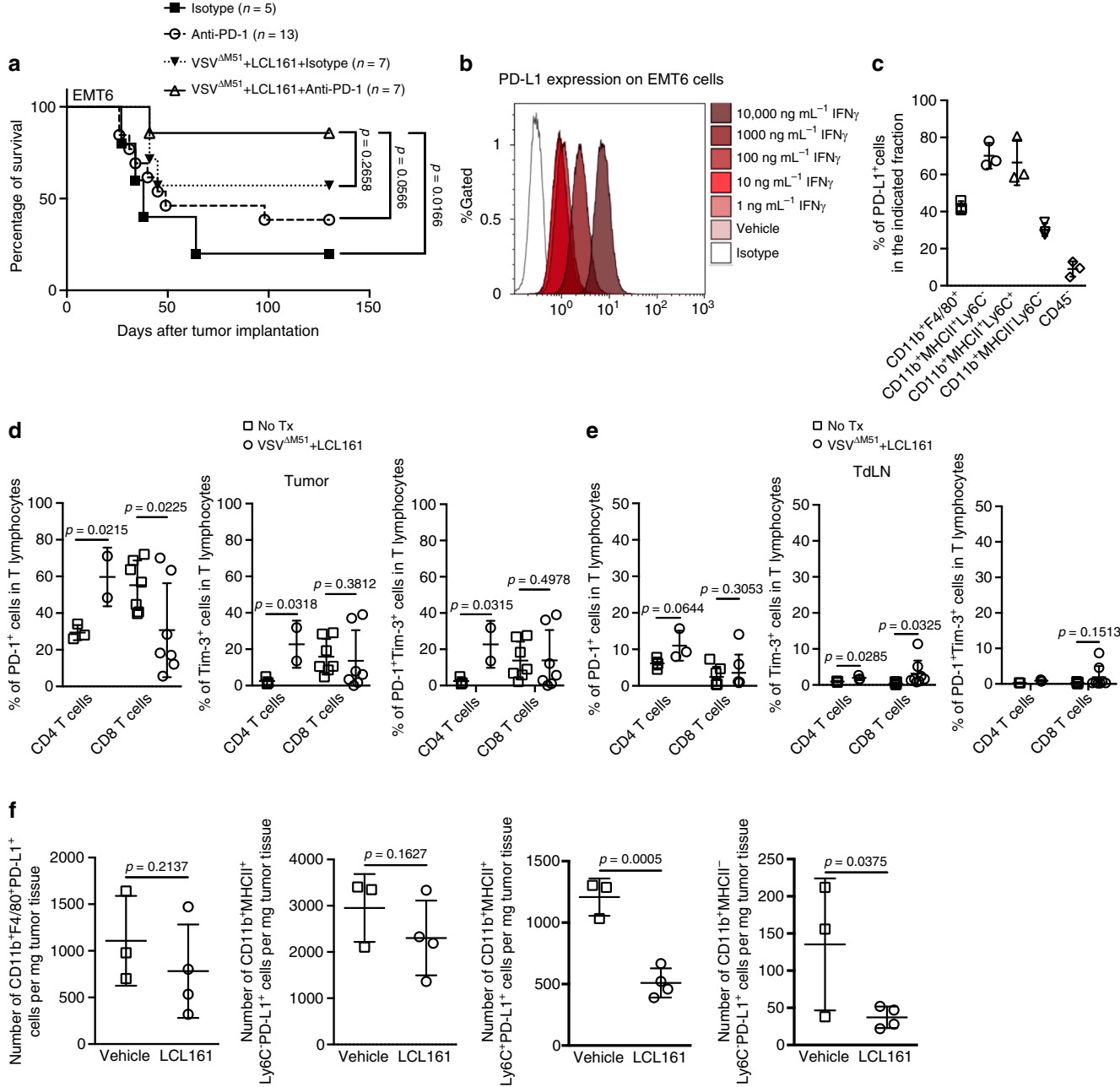

**Fig. 6** LCL161 and VSV$^{\Delta M51}$ combination therapy synergizes with αPD-1 therapy. **a** Overall survival of EMT6 tumor-bearing mice treated with LCL161 + VSV$^{\Delta M51}$ ± PD-1 antibody (or isotype control; duplicate experiments; log-rank test). **b** PD-L1 expression on EMT6 cells treated for 24 h with IFNγ (or vehicle) at the indicated concentrations, measured by flow cytometry ($n = 3$ biological replicates, single experiment). **c** PD-L1 expression on the indicated populations of cells isolated from 12-day-old EMT6 tumors (single experiment; mean ± SD). **d**, **e** Membrane staining for PD-1 and Tim-3 expression on CD8$^+$ and CD4$^+$ T cells isolated from EMT6 tumors **d** and TdLN **e** 7 days post treatment, measured by flow cytometry (duplicate experiments; mean ± SD; $t$-test). **f** PD-L1 expression on the indicated populations of TAM isolated from EMT6 tumors 72 h post TX (single experiment; mean ± SD; $t$-test)

and clinical implementation of the SMC drug class through the lens of immunotherapy should unleash its full anticancer potential.

## Methods

**Cell lines and cell culture**. EMT6 (CRL2755), 4T1 (CRL2539), CT26.WT (CRL 2638), Renca (CRL2947), Neuro2a (CCL131), RD (CCL136), Saos-2 (HTB85), MG63 (CRL1427), HOS (CRL1543), and Vero (CCL81) cells were purchased from the American Type Culture Collection (Manassas, VA, USA). TC-32, TC-71, CHLA-10, CHLA-258, Rh-30, and Rh-41 cells were obtained from the Children's Oncology Group at Texas Tech University (Lubbock, TX, USA). M3-9-M and 76–9 cells were obtained from Crystal MacKall (Stanford, CA, USA), HGF-116 cells from Timothy Cripe (Nationwide Children's Hospital, OH, USA), and CT5.3-

hTERT from Stephen Robbins (University of Calgary, AB, Canada). 4T1, CT26.WT, Renca, Neuro2a, 76–9, M3-9-M, Rh-30, Rh-41, and Tc-32 cells were propagated in RPMI 1640 (Life Technologies, Carlsbad, CA, USA) supplemented with 10% fetal bovine serum (FBS, Life Technologies). HGF-116 cells were cultured in RPMI 1640 with 15% FBS and EMT6 cells in Weymouth's medium (Life Technologies) supplemented with 15% FBS. Neuro2a, MG63, and HOS cells were cultured in Eagle's minimal essential medium (EMEM, Lonza, Basel, Switzerland) with 10% FBS and CHLA-258, CHLA-10, and Tc-71 cells were propagated in Iscove's Modified Dulbecco's Medium (IMDM, Life Technologies) with 20% FBS. Saos-2 cells were cultured in McCoy's 5A medium (Life Technologies) with 15% FBS and the RD cell line was cultured in Dulbecco's Modified Eagle Medium (DMEM, Life Technologies) with 10% FBS. Antibiotics were not added to media used for culturing cells and all lines routinely tested negative for mycoplasma. No cell line used is listed in the ICLAC database of

commonly misidentified cell lines, but they were not genetically authenticated within our lab.

**Production and purification of VSV$^{\Delta M51}$**. VSV$^{\Delta M51}$ virus (Indiana strain) was obtained from Dr David Stojdl (CHEO Research Institute)[12]. VSV$^{\Delta M51}$ was grown in Vero cells and the supernatant harvested at ~50% cytopathic effect. Cellular debris was centrifuged at 300×g for 5 min at 4 °C and virus-containing supernatant collected and passed through a 0.2 µm filter (VWR, Radnor, PA, USA). Clarified supernatant was in turn centrifuged at 28,000×g for 1.5 h at 4 °C and the pellet resuspended (1 mM EDTA, 1 mM NaCl, 1 mM Tris, pH 7.4). The virus was then centrifuged through an Optiprep gradient (MilliporeSigma, Darmstadt, Germany) at 160,000×g for 1.5 h at 4 °C. A single band of concentrated virus particles were collected, aliquoted, titered by plaque assay on Vero cells, and stored at −80 °C until use.

**Mouse tumor models**. Female BALB/c (H-2d) and C57Bl/6 (H-2b) mice (6–8 weeks old) were obtained from the Charles River Laboratory Inc. (Montreal, Canada) and maintained under specific pathogen-free conditions at the University of Calgary. For the EMT6 (including TNFR1 knockout clones) and 4T1 breast tumor models, $1–2 \times 10^5$ cells were resuspended in cold phosphate buffer saline (PBS) and injected into the fourth breast fat pad using an insulin syringe. For 76–9 and M3-9-M rhabdomyosarcoma tumor models, $1 \times 10^5$ and $1 \times 10^4$ cells, respectively, were resuspended in cold PBS and injected into the gastrocnemius muscle. Tumors were measured twice weekly using skin calipers. Tumor volume was calculated as $(\pi \times \text{length} \times \text{width}^2)/6$, where length represents the largest tumor diameter and width represents the perpendicular tumor diameter. Treatments were initiated when average tumor volume reached 100–120 mm$^3$, which occurred ~10–12 days after tumor implantation. Mice were randomly assigned to treatment group based on tumor size at treatment initiation, to ensure all groups had a similar distribution of tumor size (mean ± SD). Animal experiments were approved by The University of Calgary Conjoint Health Research Ethics Board.

**Mouse treatment regimens**. For all experiments (except Fig. 5), mice were administered $1 \times 10^8$ plaque-forming units (PFU) VSV$^{\Delta M51}$ intravenously via tail vein (resuspended in PBS), followed 6 h later by 50 mg kg$^{-1}$ LCL161 (A-1147, Active Biochem, Hong Kong, China) resuspended in 0.03 M HCl and 0.07 M NaOAc (pH 4.63) and delivered by oral gavage. Identical treatments were repeated three more times, separated by 72 h each. For experiments in Fig. 5, mice were administered two doses of LCL161 at −72 and −24 h prior to initiating the combination therapy (as described). Tumor size was measured twice weekly using skin calipers. For surgical tumor resection and rechallenge experiment, BALB/c mice were injected on the mammary fat pad with $1–2 \times 10^5$ EMT6 cells. On day 20 after tumor implantation, mice were anesthetized, the tumors completely resected, and the wound closed using silk suture. Mice with no signs of tumor growth were rechallenged with $1 \times 10^5$ EMT6 cells directly into the breast fat pad harboring the original tumor and the contra-lateral breast fat pad ~90 days after the surgical tumor resection. Tumor growth was monitored using in vivo bioluminescence imaging (Xenogen IVIS Spectrum, PerkinElmer). For antibody neutralization experiments, mice were injected intraperitoneally (i.p.) with 250 µg of either anti-CD4 (clone GK1.5; BioXcell, NH, USA) or anti-CD8 (clone 2.43; BioXcell) monoclonal antibody on day 7 after tumor implantation. Thereafter, 100 µg of either anti-CD4 or anti-CD8 were administered into the mice on days 10, 14, and 21. Control mice received rat IgG2b (BioXcell) administered by i.p. injection following the same dose and schedule. The efficiency of T-cell depletion was monitored by flow cytometry using either anti-CD4 antibody (1/100, clone RM 4–4; BioLegend) or anti-CD8 antibody (1/100, clone 53–6.7, BD Pharmingen). To determine the effect of PD-1 blockade combined with VSV$^{\Delta M51}$ and LCL161 co-therapy, mice were given four doses of i.p.-injected anti-PD-1 (250 µg per mouse, clone RMP1-14, BioXcell) separated by 72 h, beginning 1 day before the combination treatment with VSV$^{\Delta M51}$ and LCL161. Control mice were injected i.p. with rat IgG2a (BioXcell) following the same dose and schedule.

**Depletion of TAM by using CLs**. Clodronate-encapsulated liposomes (Clodrolip; 18 mg mL$^{-1}$) and empty liposome controls were provided by Dr Frank R. Jirik (University of Calgary). On day 12 after tumor implantation, EMT6 tumor-bearing mice were injected intratumorally (i.t.) with Clodrolip (1 mg per mouse) or empty liposome (60 µl per mouse) 6 h after treatment with either VSV$^{\Delta M51}$ or LCL161. Identical treatments were repeated three more times every 3 days. To test depletion of macrophages, EMT6 tumors were collected 1 day after the third treatment with Clodrolip, processed, and measured by flow cytometry.

**Immunoblotting**. Cell lysates were collected in total lysis buffer (50 mM Tris-HCl (pH 8.0), 150 mM NaCl, 1% Triton X-100, and 1% SDS). Tumor homogenates were collected by carefully dissecting out tumors by homogenizing and lysing the homogenates with tumor lysis buffer (20 mM Tris-HCl, pH 7.4, 137 mM NaCl, 2 mM EDTA, 1% Triton, and 10% glycerol) for 30 min at 4 °C. After centrifugation for 10 min at 10,000 rpm, supernatant was collected and protein quantitation performed using the Bio-Rad DC Protein Assay Kit (Bio-Rad, NC, USA). Proteins were resolved through an 8% SDS-PAGE gel and transferred to a nitrocellulose membrane using the Bio-Rad Trans-turbo semi-dry transfer apparatus at 25 V/1 A for 1 h. Membranes

were blocked with Tris-buffered-saline-Tween-20 (TBS-T) containing 5% (w/v) skim milk for 30 min at room temperature (RT) and probed overnight at 4 °C for cIAP1 and cIAP2 using either 1:3000 or 1:5000 cIAP1/2 rabbit polyclonal antibody (CY-P1041, MBL, Nagoya, Japan) for cell lysates or tumor homogenates, respectively. 1:5000 β-actin mouse monoclonal antibody (MAB1501, MilliporeSigma) was used as a loading control. The following day, membranes were washed with TBS-T three times and then probed with goat anti-rabbit (1706515, Bio-Rad, Hercules, CA) or goat anti-mouse (1706516, Bio-Rad) horseradish peroxidase-conjugated IgG for 1 h at RT. All secondary antibodies were diluted 1:5000 in TBS-T containing 5% (w/v) skim milk. Membranes were washed with TBS-T and immunoreactive proteins were detected using the Clarity™ Western ECL Substrate (Bio-Rad) on a Chemidoc-IT Imager (UVP, Upland, CA, USA).

**Enzyme-linked immunospot (ELISpot) assay**. Ninety-six-well ELISpot plates were coated with mouse anti-IFNγ monoclonal antibody overnight at 4 °C as suggested by the manufacturer (BD Biosciences, San Jose, CA, USA). The following day, plates were blocked with 10% FBS-containing RPMI 1640 for 2 h at RT, while concomitantly, CD8$^+$ T cells were isolated from single cell suspensions of TdLN (inguinal nodes) and non-draining lymph nodes (inguinal nodes of naive mice) using a mouse CD8α$^+$ T-cell isolation kit (Miltenyi Biotec, Bergisch Gladbach, Germany). Purified CD8$^+$ T cells (responder cells) were then plated into the prepared ELISpot plates ($1 \times 10^5$ cells per well) and stimulated overnight with or without live EMT6 cells (stimulator cells, $2 \times 10^4$ cells per well). Plates were washed twice with deionized water, incubated 2 h with biotinylated detection antibody, washed three times with 1× PBS containing 0.05% (v/v) Tween-20, incubated with streptavidin-HRP, and finally washed twice with 1× PBS. Spots were developed using the AEC Substrate Reagent Set (BD Biosciences) and counted on an ImmunoSpot Analyzer (Cellular Technology Ltd., Shaker Heights, OH, USA).

**Flow cytometry**. Spleens and TdLNs were isolated from mice and homogenized through a 70 µm cell strainer. The cell suspension was then centrifuged at 300×g for 5 min at 4 °C followed by red blood cell lysis using ammonium-chloride-potassium (ACK) lysis buffer (ThermoFisher Scientific, Waltham, MA, USA). Splenocytes containing white blood cells were then resuspended in 5 mL cold FACS PBS (PBS + 2% FBS, 0.01% NaN3). Cells were diluted 1:100 in FACS PBS and counted using a MOXI Z Mini Automated Cell Counter (ORFLO, Ketchum, ID, USA). A total of $1 \times 10^6$ cells were placed into an eppendorf tube, pelleted, and resuspended into 100 µL of FACS PBS. Splenocytes were blocked with 1 µL of TruStain fcX block (BioLegend) for 5 min and stained with the following antibodies for 30 min at 4 °C in the dark: FITC-conjugated anti-CD4 (1/100, clone RM 4–4, eBiosciences, San Diego, CA), PE-conjugated anti-CD8 (1/100, clone 53–6.7, BD Pharmingen), APC-conjugated anti-CD3 (1/100, clone 145-2c11, eBiosciences), APC Cy7-conjugated anti-CD45 (1/100, clone 30-F11, eBiosciences), FITC-conjugated anti-PD-1 (1/200, clone 29F.1A12, BioLegend), and PE-conjugated anti-TIM-3 (1/100, RMT3-23, eBioscience) for T lymphocytes; FITC-conjugated anti-CD11b (1/100, clone M1/70, BD Pharmingen), APC-conjugated anti-F4/80 (1/100, clone BM8, eBiosciences), PE-conjugated anti-CD169 (1/100, clone 3D6.112, BioLegend), and APC Cy7-conjugated anti-CD45 (1/100, clone 30-F11, eBiosciences) for macrophages. Following staining, cells were washed 3× with FACS PBS and quantified using the Attune Flow Cytometer (Life Technologies).

Single cell suspensions from tumors were obtained using a mouse tumor dissociation kit according to the manufacturer's instructions (Miltenyi Biotec). Briefly, tumors were isolated from mice and minced into pieces ~2–4 mm in diameter using a sterile scalpel at RT. Tumor pieces were then placed into a gentleMACS C tube (Miltenyi Biotech) with 2 mL RPMI 1640 media containing enzyme mix as provided in the kit. The samples were briefly homogenized for 1 min and incubated for 40 min at 37 °C. The samples were homogenized again in a second gentleMACS C tube for 2 min and then filtered through a 70 µm cell strainer. These single cell suspensions were centrifuged at 800×g for 10 min and the pellets were washed twice with FACS PBS, resuspended in 5 mL FACS media and enumerated using a MOXI Z Mini Automated Cell Counter. Cells were then blocked using TruStain fcX block, stained (same as above) with the appropriate antibodies (same as above), washed three times with FACS PBS, and antibody stained cells were quantified using the Attune Flow Cytometer. The absolute number of T cells or macrophages in tumors was calculated as follows: [total tumor − infiltrating immune cell count (cells per 100 mg of tumor) × percent T cells or macrophages]/100. The absolute positive cell numbers were obtained using the total cell numbers in the lymph nodes by the following formula: Absolute positive cell numbers = total cell number in the lymph nodes × percentage of positive cells × 1/100 (as in ref. [42]).

To measure the expression level of PD-L1 on EMT6 cells, cells were blocked with FcX block (1/100) for 5 min at 4 °C and were stained with Brilliant Violet 421™-conjugated anti-mouse PD-L1 (1/100, clone 10 F.9G2, BioLegend) for 30 min at 4 °C in the dark. Cells were then washed with FACS PBS twice and run on BD Attune flow cytometer. Analysis was done using Kaluza (Beckman Coulter).

**Intracellular staining**. Intracellular IFNγ and TNFα (for T lymphocytes) or arginase-1 (for macrophages) staining was conducted using the Cytofix/Cytoperm plus kit (BD PharMinigen) following the manufacturer's instructions. Briefly,

single cell suspensions from tumors or TdLNs (as described above) were seeded into 96-well round-bottomed microtiter plates ($5 \times 10^5$ cells per well) and incubated with 10 ng mL$^{-1}$ phorbol 12-myristate 13-acetate (PMA) dissolved in ethanol and 1 µg mL$^{-1}$ ionomycin dissolved in DMSO (stimulation) or vehicle control (no stimulation) for 8 h. For arginase-1 staining, the single cell activation using PMA and ionomycin was not conducted. Cells were in turn cultured with GolgiPlug (Brefeldin A, diluted 1:1000) for 8 h to retain cytoplasmic cytokines, pelleted by centrifugation, washed twice with FACS PBS, treated with TruStain fcX block (BioLegend) for 30 min on ice, and finally incubated with the following antibodies for 30 min at 4 °C in the dark: FITC-conjugated anti-CD4 (1/100, clone RM 4–4, eBiosciences), PE-conjugated anti-CD8 (1/100, clone 53–6.7 m BD Pharmingen), APC-conjugated anti-CD3 (1/100, clone 145-2c11, eBiosciences), and APC Cy7-conjugated anti-CD45 (1/100, clone 30-F11, eBiosciences) for T lymphocytes; PerCP Cy.5,.5-conjugated anti-Ly6c (1/100, clone HK1.4, eBiosciences), APC-eFluor® 780-conjugated anti-MHC II (I-A/I-E, 1/100, clone M5-114.15.2, eBiosciences), PE Cy7-conjugated anti-CD11b (1/100, clone M1/70, eBiosciences), APC-conjugated anti-CD45 (1/100, clone 30-F11, eBiosciences), PE Cy7-conjugated anti-PD-L1 (1/100, clone 10F.9G2, BioLegend), Alexa Fluor® 488-conjugated anti-F4/80 (1/100, clone BM8, eBiosciences) for macrophages. Cells were washed twice with FACS buffer, fixed, and permeabilized with Fix/Perm solution for 20 min at 4 °C in the dark. After this, cells were intracellularly stained with FITC-conjugated anti-IFNγ (1/100, clone XMG1.2, BD PharMingen) and PE-conjugated anti-TNFα (1/100, clone MP6-XT22, BD Pharmingen) for T lymphocytes; FITC-conjugated anti-arginase-1 (10 µL per $10^6$ cells, R&D systems) for macrophages. Finally, cells were washed twice with Perm/Wash solution and resuspended in 250 µL of FACS PBS for flow cytometry analysis.

To stain mouse regulatory T cells, the Mouse Foxp3 Buffer Set (BD PharMinigen) was used following the manufacturer's protocol (very similar to the instructions mentioned above). For extracellular marker staining, PE Cy7-conjugated anti-CD4 (1/100, clone RM4-4, BioLegend), PE-conjugated anti-CD25 (1/100, clone pC61.5 eBiosciences), APC-conjugated anti-CD3, (1/100, clone 145-2c11, eBiosciences), and APC Cy7-conjugated anti-CD45 (1/100, clone 30-F11, eBiosciences) were used. To stain Foxp3 intracellularly, Alexa Fluor® 488-conjugated anti-Foxp3 (1/100, clone MF23, BD Biosciences) was used after cell fixation and permeabilization. After staining, cells were analyzed using the Attune Flow Cytometer.

**Effect of LCL161 on CD8$^+$ T cells in vitro.** Single cell suspension was generated from 12-day-old EMT6 tumors and CD8$^+$ T cells were purified using a mouse CD8α$^+$ T-cell isolation kit (Miltenyi Biotec, Bergisch Gladbach, Germany) following the manufacturer's protocol. Purified CD8$^+$ T cells were counted, plated into 96-well plates ($1 \times 10^5$ cells per well), and treated for 24 h with LCL161 at various concentrations. Thereafter, cells were washed with 1× PBS twice and were stimulated with or without PMA (10 ng mL$^{-1}$) and ionomycin (1 µg mL$^{-1}$) in the presence of GolgiPlug for 8 h. Following stimulation, cells were stained for extracellular markers (CD45, CD3, and CD8), fixed, and permeabilized as mentioned above. Finally, cells were intracellularly stained with anti-IFNγ and anti-TNFα and were monitored by flow cytometry.

**Generation of EMT6-Tnfrsf1a knockout cell lines using Crispr/Cas9.** The Crispr/Cas9 system described by Sabatini and Lander[43] was used to generate functional knockouts of the tnfrsf1a locus in EMT6 cells. Cas9-EMT6 cells with doxycycline-inducible FLAG-pCas9 were generated by co-transfecting HEK293T cells with pCW-Cas9 (Addgene #50661) along with lentiviral packaging vectors psPAX2 (Addgene #12260) and pMD2.G (Addgene #12259) using Lipofectamine 2000 (ThermoFisher Scientific). Lentiviral particles were collected from the supernatant after 72 h, filtered through a 0.45 µM filter and used to infect EMT6 cells in 10 cm plates. Twenty-four hour after infection, virus was removed and media containing puromycin added. Following 7 days of selection, cells were clonally sorted using cloning rings into six-well plates. Individual clones were analyzed by western blotting for their level of FLAG-Cas9 expression in the presence or absence of 1 ug mL$^{-1}$ doxycycline and one clone chosen for generation of knockout cell lines on the basis of its robust induction of FLAG-Cas9 in the presence of doxycycline and low-level "leakiness" in its absence.

Guide sequences that target the tnfrsf1a locus were derived from a published list of Crispr sgRNAs for the mouse genome[44]. Target sequences were cloned into the pLX-sgRNA vector (Addgene #50662) using overlap-extension PCR to generate sgRNA-specific inserts[5]. Briefly, PCR amplicons produced from F1/R1 and F2/R2 primer pairs were gel purified, mixed, and used as template for PCR with the F1/R2 primer pair. The resulting product was digested along with pLX-sgRNA using NheI and XhoI, ligated and transformed into DH5α bacteria.

Lentiviral particles encoding target sgRNAs were produced by co-transfecting HEK293T cells with the pLX-sgRNA targeting constructs (sgRNA1, sgRNA2, and sgRNA3) along with lentiviral packaging vectors psPAX2 and pMD2.G using Lipofectamine 2000, as described above. Collected lentiviral particles were subsequently used to infect Cas9-EMT6 cells in 10 cm plates and following 2 weeks blasticidin selection, Cas9 expression was induced by addition of 1 µg mL$^{-1}$ doxycycline to the media. Cells were then clonally sorted into six-well plates using cloning rings and analyzed functionally for their sensitivity to TNFα, as described below. Genomic edits in clones were identified by PCR and Sanger sequencing. An

891 bp and a 1554 bp region surrounding the sgRNA target sites were amplified from genomic DNA using primer pairs tnfrsf1a-for3/tnfrsf1a-rev5 and tnfrsf1a-for3/tnfrsf1a-rev1, respectively. PCR products were generated using Taq polymerase to produce T-overhangs and subcloned into the Topo-A cloning vector to allow for allelic analyses (ThermoFisher Scientific). Subcloned amplicons were sequenced with standard M13F and M13R-17 primers. Primers for genomic analyses of clones are shown as follows: Tnfrsf1a-for3, 5′-CGGCTTCTT TTGCTTGTTTC-3′; Tnfrsf1a-rev1, 5′-AGGTAAGAACTTGCCCAAGG-3′; Tnfrsf1a-rev5, 5′-CTTACCTGTGGGAAAGCGGT-3′.

**Production of conditioned media.** EMT6 cells were infected with VSV$^{\Delta M51}$ at a multiplicity of infection (MOI) of 0.1 for 24 h. The cell culture supernatants were collected and irradiated for 12 min with maximum energy ultra violet light using a Stratalinker 1800 UV crosslinker (Stratagene, San Diego, CA, USA) to inactive VSV$^{\Delta M51}$ particles. Following centrifugation at 1000×g for 5 min at 4 °C, supernatants were apportioned into 5 mL aliquots and kept at −80 °C until use.

**In vitro cell viability assays.** Various mouse cancer cell lines (Supplementary Fig. 5), human pediatric sarcoma lines (Supplementary Fig. 7), and EMT6 clones edited by Crispr-Cas9 (Fig. 1c) were used for in vitro cell viability assays. A total of $1.0–2.5 \times 10^3$ cells per well were seeded in 96-well plates overnight and then treated with DMSO or LCL161 at 100 nM for 2 h in 50 µL of media. Following this, 50 µL of media alone, media containing 2–200 ng mL$^{-1}$ of either mouse or human recombinant TNFα, or virus conditioned media, was added. Cells were then incubated for 72 h and an alamarBlue® assay (ThermoFisher Scientific) was performed according to the manufacturer's protocol. The absorbance of each sample was then measured at 570 and 600 nm using a SpectraMax i3 spectrophotometer (Molecular Devices, Sunnyvale, CA, USA). Data were expressed as percent viability compared to control per untreated group.

$$\% \text{ Viability} = \frac{\text{Abs} (570 - 600 \text{ nm}) \text{ sample}}{\text{Abs} (570 - 600 \text{ nm}) \text{ control}} \times 100.$$

To neutralize TNFα signaling in vitro, 10 µg mL$^{-1}$ of anti-mouse TNFα (AF-410-NA, R&D Systems, Minneapolis, MN, USA) or isotype control (normal goat IgG, AB-108-C, R&D Systems) was added to EMT6 culture media for 2 h before LCL161 treatment followed 2 h later by mouse TNFα or conditioned media as mentioned above.

BMDMs were seeded in 24-well tissue culture plates at $2 \times 10^5$ BMDMs per well and treated with LCL161 or vehicle (DMSO) for 24 h at 37 °C. Where indicated, cells were also treated with TNF at 100 ng mL$^{-1}$. Each condition was assayed in duplicate for lactate dehydrogenase (LDH) release using a Pierce LDH Cytotoxicity Assay Kit (Thermo Scientific) as per the manufacturer's protocol. Using an Envision 2104 Multilabel Reader (PerkinElmer), absorbance at 490 nm was measured five times and averaged, and average absorbance at 680 nm (background) was subtracted to determine final values. Results were made relative to maximum LDH release as determined by BMDMs treated with 1× lysis buffer (Thermo Scientific) for 45 min prior to measuring LDH activity.

**Sample collection for cytokine analyses.** To collect tumor interstitial fluid (TIF), 0.1–0.3 g of fresh tumor tissues was cut into small pieces (1–3 mm$^3$), washed in 2 mL of 1× cold PBS, and placed in a 15 mL conical plastic tube containing 1× cold PBS (0.25 g tumor tissue per 1 mL PBS). Samples were then incubated for 1 h at 37 °C in a humidified CO$_2$ incubator, centrifuged at 1000 rpm for 3 min, and the supernatants transferred to new microtubes. Following an additional centrifugation at 5000 rpm for 20 min at 4 °C, supernatants were collected, aliquoted, and frozen at −20 °C.

To obtain serum, mice were anesthetized with a ketamine and xylazine cocktail (100 and 10 mg kg$^{-1}$, respectively) and blood was drawn by cardiac puncture. Whole blood was clotted for 30–60 min then centrifuged at 1000–2000×g for 20 min. Serum was apportioned into 0.1–0.3 mL aliquots and kept at −80 °C.

To check the level of cytokines produced in cultured cells, culture media was collected from cells treated with either VSV$^{\Delta M51}$ at the indicated MOI or LCL161 (50 nM) or both, at indicated time points, centrifuged at 1000×g for 5 min and kept at −80 °C until analysis.

**Cytokine analysis.** TIF, serum, and tissue culture supernatants were sent to Eve Technologies or the Snyder Institute Translational Laboratory in Critical Care Medicine (University of Calgary) for analysis via Multiplex ELISA (Luminex). IFNα or IFNβ in TIF, serum, or tissue culture supernatants were measured using a Verikine mouse IFNα or IFNβ ELISA kit, respectively, according to the manufacturer's instructions (PBL Assay Science, Piscataway, NJ, USA).

**Quantitative RT-PCR.** EMT6 tumors were homogenized with a mortar and pestle followed by RNA extraction using the RNeasy® plus mini kit (Qiagen). As a cellular control, EMT6-fluc RNA was isolated 8 h post infection with VSV$^{\Delta 51}$ at MOI 1. An aliquot of 500 ng of total RNA for each sample was used for cDNA synthesis

with the High Capacity cDNA Reverse Transcription kit (Applied Biosystems). Semi-quantitative RT-PCR amplification was performed in triplicates using iQ™ SYBR® Green Supermix (Bio-Rad) and 600 nM of gene-specific primers (i.e., *GAPDH*, *IFNα*, *IFNβ*, *IRF7*, *OAS*, *STAT1*, *Mx2*, *iNOS*, and *Arg1*) with the CFX96 Real-Time system. Initial PCR denaturation was at 95 °C for 2 min followed by 40 cycles of 95 °C for 15 s, 55–63.3 °C for 20 s, and 72 °C for 20 s. Primer annealing temperature varied with each primer set (Supplementary Table 1). A melt curve was performed from 55 to 95 °C, with 0.5 °C increments every 5 s. The mRNA expression was normalized to the respective *GAPDH* levels of each sample.

**Purification and culture of BMDMs**. The lower half of a mouse leg, including femur, ilium, and tibia as well as the surrounding musculature, was removed and transferred to a tissue culture hood. Muscles and bones were placed in 70% ethanol for 10 s, flamed briefly to sterilize the tissue and placed into DMEM + 10% FBS. Muscles were removed using forceps and a scalpel to expose the bones, the ends of which were then cut with sterile scissors. Cut bones were flushed with 20 mL of DMEM + 10% FBS using a 21G needle and bone marrow cells were made into single cell suspensions by gently pipetting them with a serological pipette. Cells were enumerated with a Moxi Z cell counter (Orflo), centrifuged at 300×g for 5 min at 4 °C and cultured with the appropriate medium at 37 °C.

Macrophages were derived from the above preparation by first removing red blood cells using sterile ACK lysis buffer, then washing in cold PBS and resuspending in DMEM supplemented with 10% FBS, 2 mM L-glutamine, 1 mM sodium pyruvate, 20% L-cell-conditioned media containing macrophage colony-stimulating factor, and 1× penicillin-streptomycin antibiotic cocktail (ThermoFisher Scientific). Cells were counted and maintained at a concentration of $2 \times 10^6$ cells mL$^{-1}$ with media changes every 3 days. Differentiated BMDMs were obtained after ~7 days of culture.

**M1/M2 macrophage activation assays**. BMDMs were seeded in 24-well tissue culture plates at $2 \times 10^5$ BMDMs per well and activated toward M1-like or M2-like with 10 ng mL$^{-1}$ IFNγ or IL-4 (Peprotech), respectively, for 40 h at 37 °C alongside simultaneous treatment with LCL161 or vehicle (DMSO). BMDMs incubated in media without IFNγ or IL-4 were used as a negative control. After 40 h, cells were processed for qPCR analysis of arginase-1 or flow cytometry analysis of iNOS, as described.

**ROS production during phagocytosis in BMDM**. BMDMs were seeded into 96-well tissue culture plates (Greiner Bio-One, Monroe, NC, USA) at $1 \times 10^5$ BMDMs per well and pre-treated with 100 U mL$^{-1}$ IFNγ (100 ng mL$^{-1}$), LCL161, or DMSO for 20 h. To measure the release of $H_2O_2$ into the supernatant, BMDM monolayers were washed and then incubated for 1 h in assay buffer (tissue culture grade PBS supplemented with 1 mM CaCl$_2$, 2.7 mM KCl, 0.5 mM MgCl$_2$, 5 mM dextrose, and 0.25% gelatin) containing 10 mg mL$^{-1}$ serum-opsonized zymosan as previously described[7]. Amplex UltraRed (Life Technologies), at a final concentration of 10 ng mL$^{-1}$ plus 1 unit horseradish peroxidase (Sigma-Aldrich), was added to each well supernatant post-zymosan exposure and incubated for 15 min. Amplex UltraRed fluorescence was monitored using a FLUOstar OPTIMA microplate reader (BMG Labtech, Ortenberg, Germany). BMDMs were treated with the NOX2 inhibitor DPI (0.5 mM) where indicated.

**Antigen presentation by BMDM**. OT-1 T cells (transgenic for a CD8$^+$ T-cell receptor specific for OVA (SIINFEKL) in the context of MHC I) or 2D2 T cells (transgenic for a CD4$^+$ T-cell receptor (Vβ11/Vα3.2) specific for myelin oligodendrocyte glycoprotein (MOG35–55) in the context of MHC II (I-Ab)) were used to assay antigen presentation efficiency by class I and II MHC, respectively. BMDMs were seeded in 96-well tissue culture plates at $1 \times 10^5$ BMDMs per well and pre-treated with IFNγ (100 ng mL$^{-1}$), LCL161, or DMSO for 20 h. BMDMs were pulsed with MOG35-55 peptide (25 μg mL$^{-1}$, University of Calgary Peptide Services, Calgary, AB, Canada) or full-length OVA (Worthington Biochemical Corporation, LS003059) for 6 h, and following extensive washing, further co-cultured with naive splenocytes from 2D2 or OT-1 mice ($5 \times 10^5$ cells per well) for 16 h. Subsequent flow cytometry analysis of CD69 surface expression (an early T-cell activation marker) on gated CD4$^+$ or CD8$^+$ T cells was used to determine the efficiency of MOG or OVA presentation by MHC-II and MHC-I, respectively.

**Measurement of BMDM activation markers induced by LCL161**. BMDMs were seeded into 96-well μClear tissue culture plates (Greiner Bio-One) at $1 \times 10^5$ BMDMs per well and treated with varying concentrations of LCL161 or DMSO for 20 h. After washing cells with FACS PBS twice, cells were blocked with FcX block (1/100) for 5 min at 4 °C and were stained for the following cell surface markers: PE-conjugated CD40L (clone MR1, BioLegend), FITC-conjugated CD80 (Clone 16-10A1, BD Biosciences), AF647-conjugated anti-MHC I (H-2Kb, clone AF6-88.5, BD Pharmingen), and APC-eFluor 780-conjugated anti-MHC II (I-A/I-E, clone M5/114.15.2, eBiosciences) for 30 min at 4 °C in the dark. After cells were washed with FACS PBS twice, the expression level of activation markers

on BMDMs was measured using BD Attune flow cytometer. Analysis was done using Kaluza (Beckman Coulter).

**Immunofluorescence**. Frozen tumor or spleen tissues embedded in optimal cutting temperature compound (Sakura Finetek, the Netherlands) were sectioned (6 μm) for immunofluorescence. After fixation with ice-cold acetone for 10 min, sections were blocked and permeabilized with 1× PBS containing 5% normal goat serum, 0.1% Triton X-100, and 1% bovine serum albumin for 30 min at RT. Sections were then incubated overnight at 4 °C with a rat monoclonal antibody to F4/80 (1:300, clone BM8, Santa Cruz Biotechnology, Dallas, TX, USA). The following day, sections were washed in 1× PBS and incubated 1 h at RT with Alexa 488 goat anti-rat secondary antibody (1:400, Life Technologies). Sections were counter stained with 300 nM DAPI (4′,6-diamidino-2-phenylindole, BioLegend) for 2–5 min at RT.

**IVM**. Mice were anesthetized by i.p. injection of 200 mg kg$^{-1}$ ketamine (Bayer Inc. Animal Health, Toronto, ON, Canada) and 10 mg kg$^{-1}$ xylazine (Bimeda-MTC, Cambridge, ON, Canada). The tail vein was cannulated to permit the delivery of fluorescently labeled antibodies (5–10 μg). Resonant-scanning confocal IVM was performed using a Leica SP8 inverted microscope (Leica Microsystems, Concord, ON, Canada), equipped with 405-, 488-, 552-, and 638-nm excitation lasers, 8 kHz tandem scan head and spectral detectors (conventional PMT and hybrid HyD detectors).

Imaging of the TdLN was facilitated by the creation of a skin flap. Briefly, midline incision along the spine was performed and skin reflected. The thin connective tissue membrane overlaying the inside surface of the skin was removed and edges of this skin flap were secured by sutures to expose and stabilize the inguinal LN imaging.

For imaging of the spleen, the organ was externalized by making a 1 cm incision in the skin and musculature at the left dorsal side of the animal and gently tethered out of the body using 3–0 sutures tied to connective tissue associated with the spleen. The mouse was then laid on a stage with the spleen positioned over a cover slip.

**Quantification of virus particles**. Tumors, TdLNs, and spleens were excised using sterile forceps, scalpel, and scissors and any remaining skin was removed from the tissues. Tissues were flash frozen in 95% ethanol and dry ice and stored at −80 °C. For analysis, tissues were thawed on ice and homogenized using a Homogenizer Model 125 (ThermoFisher Scientific) in serum-free DMEM at a concentration of 50 mg mL$^{-1}$ of DMEM. Homogenates were centrifuged at 12,000 rpm for 5 min at 4 °C, supernatants collected, and virus concentration quantified by plaque assay. Plaque assays were performed as follows: $5 \times 10^5$ Vero cells were plated in six-well plates overnight to establish a confluent monolayer of cells. The solution containing the unknown concentration of virus was serially diluted in 1:10 increments, vortexed at high speed, and 100 μl added to the Vero cells. Following incubation at 37 °C for 1 h with manual shaking every 15 min, 2 mL of media and agarose mixture (1:1, 1.2% agarose with 2× DMEM + 20% FBS + 2× penicillin-streptomycin antibiotic cocktail) was overlain using a serological pipette. Plates were then incubated at 37 °C, 5% CO$_2$ for 18–20 h and viral plaques were visualized by staining with 0.05% (w/v) crystal violet in 17% (v/v) methanol for 2 h at RT. For viral titering, the number of infectious virus particles was expressed as PFU per gram of tissue.

**Statistical analyses**. Data analysis was performed using GraphPad Prism 6.0c (GraphPad Software, La Jolla, CA, USA). Survival curves were analyzed using log-rank tests, with differences between groups tested for using the Bonferonni correction. For all other data, statistical comparison between two groups were conducted using a two-tailed, unpaired Student's *t*-test, while comparisons of more than two groups were performed using one- or two-way analysis of variance (ANOVA) with Tukey's post hoc test. In all cases, $p < 0.05$ was considered a statistically significant difference. All values are reported as mean ± SD, except qPCR data (mean ± SEM). All values and variances were generated from biological replicates. All data were tested for normality prior to statistical analysis (GraphPad).

Most experiments were replicated one–two times. For animal experiments, the sample size of the initial experiment was generally set to $n = 3–5$ per group. Replication was performed to either reproduce a statistically significant experimental result, or to add power to a result that approached statistical significance. The *n* value and number of experimental replications is defined within each figure and/or legend. A qualitatively similar approach was taken for cell culture experiments. Experiments were not blinded.

**Data availability**. Data supporting the findings of this study are available within the article and its supplementary information files or from the corresponding author on reasonable request.

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

## Acknowledgements

This research was funded by operating grants from the Cancer Research Society (CRS) and Alliance for Cancer Gene Therapy (ACGT), and philanthropic support from the Alberta Children's Hospital Foundation, the Kids Cancer Care Foundation (KCCF), and the Believe in the Gold Foundation.

## Author contributions

D.-S.K. performed the majority of experiments, analyzed the majority of data, and helped draft the manuscript. H.D., C.Z., K.L., A.R., J.R., and S.S. helped with cell culture and animal experiments. V.N. performed the IVM (supervised by C.J.). D.B., B.E., F.J.Z., and P.T. isolated BMDMs and/or performed BMDM experiments (supervised by R.Y.). M.E. and C.G. generated CRISPR/Cas9 clones. C.G. and F.J.Z. provided critical feedback on the manuscript. D.J.M. conceptualized the project, helped analyze the data, and wrote the manuscript.

## Additional information

**Competing interests:** The authors declare no competing financial interests.

