## [Peer Review File · Nature Communications]

Reviewers' comments:

Reviewer #1 (Remarks to the Author):

This manuscript extends the findings of this group, published previously in Nature Biotechnology, that VSV and SMC therapies can be combined to produce better efficacy than either alone. In the current study, the authors explore the mechanisms of the therapy. They show that the combination generates long term memory against tumor and that it is dependent upon CD8+ve T cells. Using each agent alone, they propose that the SMC therapy acts through re-invigorating exhausted T cells in the tumor and that this is due to its ability to reverse immune suppression by tumor associated macrophages. They also present data confirming that VSV is highly inflammatory and recruits T cells to the tumor.

Specific Comments:

Figure 1A/B: The authors show that mice in which tumors were put into durable remission by VSV+SMVC therapy reject subsequent challenge with EMT 6 cells. The control here was rechallenge of naïve mice. To be certain that the long term memory was due to the nature of the primary therapy (VSV+SMC) it would be better to see a control of mice which had seen EMT6 cells in another form – such as irradiated cells. This would answer the question of whether the tumor cell line is immunogenic itself (ie any exposure to these cells would vaccinate against re-challenge).

Figure 1C: Is the treatment with VSV+LCL161 + Isotype statistically significant over that with LCL161 + Isotype?

Figure 2: The authors postulate that SMC therapy reverses a state of severe T cell exhaustion for the T cells in the tumor. Their evidence for this is the inability to stimulate TIL ex vivo with PMA/ionomycin unless the mice were treated with SMC. Are these TIL expressing PD-1/TIM3 checkpoint molecules? Do the EMT6 tumors express PD-L1?

Figure 2: The role of LCL161 on depleting macrophages is postulated from a series of in vitro experiments. Have the authors attempted depletion of these TAM to show that SMC therapy mimics the role of immune suppressive TAM?

Figure 3: The role of VSV as a potent systemic immune adjuvant is not really novel. In this model, where systemic VSV has no therapy effect alone, does the virus reach the tumors? The authors look in the TDLN but do not present data of virus in the tumor itself. Could the virus be activating TIL or de-suppressing TAM in the tumor?

In summary, there is a lot of data presented to support the individual modes of action of SMC (immune stimulatory to T cells through modification of immune suppressive TAM) and VSV (generalized immune adjuvant). The advance on previous work published by the authors here is highly mechanistic and perhaps not particularly unexpected given previous knowledge of the actions of both LCL161 and VSV.

Reviewer #2 (Remarks to the Author):

The authors show that a combination of SMAC-mimetic compounds and oncolytic virus treatment work synergistically. Although this is not new as there is one previous paper on this concept, the authors made considerable important steps. In the previous report it was - based on in vitro

experiments - stated that the underlying mechanism of synergy involved TNF α triggered apoptosis. The authors of this manuscript confirm the in vitro experiments of the previous manuscript and importantly show that in vivo this mechanism does not play a major role. In vivo, other mechanisms play a more dominant role and these mechanisms are very nicely dissected in this manuscript. The experiments are well performed, the data is very convincing and almost complete. The outcomes of this manuscript are of extreme importance to the field as currently different types of combinations of immunotherapy are tested, and the rational design of such combinations can only be made on a correct understanding of the mechanisms of each compound in vivo.

There are a few questions left, the answers to which would improve the manuscript.

1. In the two mouse models used, EMT6 and M3-9-M, a CD4 T cell depletion was performed. While in EMT6 CD4 depletion improved survival of the mice, this was not the case in M3-9-M. The authors suggest that in the case of EMT6 this is due to the depletion of high numbers of regulatory T cells that infiltrate the tumor and suppress immunity. It would be nice if the authors could substantiate this by - at least - showing how the CD8, CD4 and especially Treg infiltrate of EMT6 and M3-9-M tumors differ under no-treatment circumstances.
2. Lines 90-91, Alternatively, VSV primes for better response to LCL161 (see also point 3)
3. Line 120, figure 2k: while it is stated that LCL161 treated BMDC were hyperresponsive to VSV this can not be concluded from the figure. There is no LCL161 only control shown, there is only a VSV-only control shown. Based on this one can only conclude that the increase response in cytokines is due to LCL161-induced hyperresponses after VSV treatment. This should be corrected (see also point 2).
4. Line 123, the authors claim that LCL161-treated BMDC displayed enhanced ability to present antigen (figure 2m) but this can not be concluded. They only show that the treated BMDC have a stronger capacity to activate CD4 T cells. This may be due to higher expression of antigen (Signal 1) but it might also be that co-stimulation (signal 2) or cytokine production (signal 3) is altered, leading to better T cell activation. For the latter, they have the evidence.
5. It would have been nice if the authors would have used in figure 2m CD8 T cells, since they show that their model is dependent on CD8 T-cell reactivity rather than CD4 T cells.
6. Line 125, figure S11: the authors claim that the viability of LCL161 treated BMDC is decreased. First, this should be shown in a quantitative manner, one can not really judge this by a picture. Secondly, how would this conceptually work, why activate and then kill M1 macrophages?
7. Lines 125-126: the authors state that sustained LCL161 treatment in vitro results in decreased cell number. Please explain "sustained", and how different is this from their in vivo experiments, in other words how relevant is this experiment?
8. Line 150: without altering their overall numbers, should be changed into without significantly altering
9. Lines 150-159, while the authors make a strong case for the effect of VSV to infect antigen presenting cells thereby stimulating the production of large quantities of cytokines suggesting this may work as adjuvant for T cells, it was not proven beyond doubt. Potentially, final proof could be obtained by depletion of antigen presenting cells before treatment and then measure cytokines. For instance, through clodronate treatment.

Reviewers' comments:

Reviewer #1 (Remarks to the Author):

This manuscript extends the findings of this group, published previously in *Nature Biotechnology*, that VSV and SMC therapies can be combined to produce better efficacy than either alone. In the current study, the authors explore the mechanisms of the therapy. They show that the combination generates long term memory against tumor and that it is dependent upon CD8+ve T cells. Using each agent alone, they propose that the SMC therapy acts through re-invigorating exhausted T cells in the tumor and that this is due to its ability to reverse immune suppression by tumor associated macrophages. They also present data confirming that VSV is highly inflammatory and recruits T cells to the tumor.

Specific Comments:

Figure 1A/B: The authors show that mice in which tumors were put into durable remission by VSV+SMC therapy reject subsequent challenge with EMT6 cells. The control here was rechallenge of naïve mice. To be certain that the long-term memory was due to the nature of the primary therapy (VSV+SMC) it would be better to see a control of mice which had seen EMT6 cells in another form – such as irradiated cells. This would answer the question of whether the tumor cell line is immunogenic itself (i.e., any exposure to these cells would vaccinate against re-challenge).

Figure 1C: Is the treatment with VSV+LCL161 + Isotype statistically significant over that with LCL161 + Isotype?

Figure 2: The authors postulate that SMC therapy reverses a state of severe T cell exhaustion for the T cells in the tumor. Their evidence for this is the inability to stimulate TIL ex vivo with PMA/ionomycin unless the mice were treated with SMC. Are these TIL expressing PD-1/TIM3 checkpoint molecules? Do the EMT6 tumors express PD-L1?

Figure 2: The role of LCL161 on depleting macrophages is postulated from a series of in vitro experiments. Have the authors attempted depletion of these TAM to show that SMC therapy mimics the role of immune suppressive TAM?

Figure 3: The role of VSV as a potent systemic immune adjuvant is not really novel. In this model, where systemic VSV has no therapy effect alone, does the virus reach the tumors? The authors look in the TDLN but do not present data of virus in the tumor itself. Could the virus be activating TIL or depressing TAM in the tumor?

In summary, there is a lot of data presented to support the individual modes of action of SMC (immune stimulatory to T cells through modification of immune suppressive TAM) and VSV (generalized immune adjuvant). The advance on previous work published by the authors here is highly mechanistic and perhaps not particularly unexpected given previous knowledge of the actions of both LCL161 and VSV.

Reviewer #2 (Remarks to the Author):

The authors show that a combination of SMAC-mimetic compounds and oncolytic virus treatment work synergistically. Although this is not new as there is one previous paper on this concept, the authors made considerable important steps. In the previous report it was - based on in vitro experiments - stated that the underlying mechanism of synergy involved TNF α triggered apoptosis. The authors of this manuscript confirm the in vitro experiments of the previous manuscript and importantly show that in vivo this mechanism does not play a major role. In vivo, other mechanisms play a more dominant role and these mechanisms are very nicely dissected in this manuscript. The experiments are well performed, the data is very convincing and almost complete. The outcomes of this manuscript are of extreme importance to the field as currently different types of combinations of immunotherapy are tested, and the rational design of such combinations can only be made on a correct understanding of the mechanisms of each compound in vivo.

There are a few questions left, the answers to which would improve the manuscript.

1. In the two mouse models used, EMT6 and M3-9-M, a CD4 T cell depletion was performed. While in EMT6 CD4 depletion improved survival of the mice, this was not the case in M3-9-M. The authors suggest that in the case of EMT6 this is due to the depletion of high numbers of regulatory T cells that infiltrate the tumor and suppress immunity. It would be nice if the authors could substantiate this by - at least - showing how the CD8, CD4 and especially Treg infiltrate of EMT6 and M3-9-M tumors differ under no-treatment circumstances.
2. Lines 90-91, Alternatively, VSV primes for better response to LCL161 (see also point 3)
3. Line 120, figure 2k: while it is stated that LCL161 treated BMDC were hyperresponsive to VSV this cannot be concluded from the figure. There is no LCL161 only control shown, there is only a VSV-only control shown. Based on this one can only conclude that the increase response in cytokines is due to LCL161-induced hyper-responses after VSV treatment. This should be corrected (see also point 2).
4. Line 123, the authors claim that LCL161-treated BMDC displayed enhanced ability to present antigen (figure 2m) but this cannot be concluded. They only show that the treated BMDC have a stronger capacity to activate CD4 T cells. This may be due to higher expression of antigen (Signal 1) but it might also be that co-stimulation (signal 2) or cytokine production (signal 3) is altered, leading to better T cell activation. For the latter, they have the evidence.
5. It would have been nice if the authors would have used in figure 2m CD8 T cells, since they show that their model is dependent on CD8 T-cell reactivity rather than CD4 T cells.
6. Line 125, figure S11: the authors claim that the viability of LCL161 treated BMDC is decreased. First, this should be shown in a quantitative manner, one cannot really judge this by a picture. Secondly, how would this conceptually work, why activate and then kill M1 macrophages?
7. Lines 125-126: the authors state that sustained LCL161 treatment in vitro results in decreased cell number. Please explain "sustained", and how different is this from their in vivo experiments, in other words how relevant is this experiment?
8. Line 150: without altering their overall numbers, should be changed into without significantly altering

9. Lines 150-159, while the authors make a strong case for the effect of VSV to infect antigen presenting cells thereby stimulating the production of large quantities of cytokines suggesting this may work as adjuvant for T cells, it was not proven beyond doubt. Potentially, final proof could be obtained by depletion of antigen presenting cells before treatment and then measure cytokines. For instance, through clodronate treatment.

Summary of changes made to the manuscripts:

The experiments requested by the reviewers led to the generation of significant new data. We have added these data to our original data, and reorganized the manuscript considerably, which now consists of 6 primary figures and 21 supplemental figures.

1. Figure 1:

- a. Removed original panels a-b.
- b. Merged original panels k-l (now panel i).

2. Figure 2:

- a. Added new data in panel b,e (PD-1 and Tim-3 expression on CD8 TIL and TdLN)
- b. Added new data in panel h (direct stimulation of isolated CD8 T cells with LCL161)
- c. Moved panels g-m to Figure 3 (a-d).
- d. Removed panel n and moved panel o to supplemental figure 15

3. Figure 3:

- a. Original panels 3a-f moved to Figure 4.
- b. Original panels 3g-j moved to Figure 5.
- c. New Figure 3 now comprises Original Figure 2g-m (now Fig. 3a-d), new data in 3e (Arg-1 expression on TAM), original figures 2j-m (now Fig. 3f-g,i) and new data in h, j-k (BMDM activation/polarization data)

4. Figure 4:

- a. Panels 4a-c are original panels 3a-c.
- b. Panel 4d are new data (Arg-1 expression in TAM).
- c. Panels 4e-g are original panels d-f.

5. Figure 5:

- a. Panel a is from original Fig. 3g, plus with the addition of a new duplicated experiment.
- b. Panel b is from original Fig. 3h.
- c. Panel c is from original Fig. S15.
- d. Panel d is from original Fig. 3j, plus the addition of new tumour and TdLN data

6. Figure 6: Entirely new data

7. Figure S1-5: as in original manuscript

8. Figure S6: new data

9. Figures S7: original S6

10. Figure S8: new data

11. Figure S9: original S7

12. Figure S10: new data

13. Figures S11-13: original S8-10

14. Figure S14: original S11, plus new quantitative data

15. Figure S15-18: new data

16. Figure S19-21: original S12-14

Authors' point-by-point response:

Summary of remarks by Reviewer #1: This referee is in agreement with the main points of our manuscript, that our data “*support the individual modes of action of SMC (immune stimulatory to T cells through modification of immune suppressive TAM) and VSV (generalized adjuvant)*”. A number of experiments were requested, generally aimed toward providing additional support for/insight into the proposed mechanism(s) of action.

1. Figure 1A/B: The authors show that mice in which tumors were put into durable remission by VSV+SMC therapy reject subsequent challenge with EMT6 cells. The control here was rechallenge of naïve mice. To be certain that the long-term memory was due to the nature of the primary therapy (VSV+SMC) it would be better to see a control of mice which had seen EMT6 cells in another form – such as irradiated cells. This would answer the question of whether the tumor cell line is immunogenic itself (i.e., any exposure to these cells would vaccinate against re-challenge).

Authors' response: This is an excellent point. We opened our original manuscript with these experiments as a prelude to the much more incisive T cell depletion experiments (for determining the role of T cells in the therapeutic response). Although we didn't discuss the possibility that those data may reflect a strong innate immunogenicity within EMT6 cells, as suggested by Reviewer #1, we did propose that a defining feature of tumour model sensitivity to SMC±VSV immunotherapy is immunogenicity (**original Figs. 1i-k, Fig. S5c**). Because others have already shown that EMT6 tumours are immunogenic through irradiated cell vaccination experiments¹, we decided to address this comment by surgically resecting established EMT6 tumours and asking whether those Balb/c mice accepted or rejected an EMT6 graft 90 days later. As shown in **Fig. S6**, 100% of mice cured of EMT6 tumours by surgical resection rejected subsequent engraftment of EMT6 cells. This result is consistent with the irradiated cell vaccine experiment in demonstrating that EMT6 cells are sufficiently immunogenic to establish a long-lasting immunity in Balb/c mice in the absence of any treatment.

We do not suggest that the SMC±VSV immunotherapy engenders *de novo* immunity toward the tumour, but rather that it changes the conditions within the TME and TdLN in several ways that ultimately allow for more effective, pre-existing immune responses against the cancer. In light of this comment by Reviewer #1, we opted to remove **original Fig. 1a-b** from the revised manuscript, given that the T cell depletion experiment is a much better experiment anyway.

2. Figure 1C: Is the treatment with VSV+LCL161 + Isotype statistically significant over that with LCL161 + Isotype?

Authors' response: This experiment was designed to test whether CD8⁺ cell depletion affected outcomes to LCL161±VSV therapy. While sufficiently powered to detect a CD8⁺ cell depletion effect, it was underpowered to detect a statistically significant difference between LCL161 (n=3) and LCL161+VSV (n=3). Nevertheless, the usual trend was observed (i.e., LCL161+VSV is more effective than LCL161 alone). When sufficiently powered (see ref² or **Supplemental Fig. 1a**) a statistically significant difference between these two treatment groups is consistently and robustly observed.

3. Figure 2: The authors postulate that SMC therapy reverses a state of severe T cell exhaustion for the T cells in the tumor. Their evidence for this is the inability to stimulate TIL ex vivo with PMA/ionomycin unless the mice were treated with SMC. Are these TIL expressing PD-1/TIM3 checkpoint molecules? Do the EMT6 tumors express PD-L1?

Authors' response: We performed several experiments to address these comments:

1. We isolated T cells from established EMT6 tumours and draining lymph node and performed flow cytometry for PD-1 and Tim-3 surface marker expression. For CD8⁺ TIL, we found that ~70% were PD-1 positive and ~30% were Tim-3 positive or double positive, which provides molecular confirmation of our functional analyses of TIL exhaustion (**Fig. 2b**). In contrast, less than 5% of CD8⁺ cells in the TdLN were positive for PD-1 and/or Tim-3, again relatively consistent with our functional studies (**Fig. 2e**). Qualitatively similar results were obtained from CD4⁺ cells, although quantitatively a smaller percentage of CD4⁺ TIL were PD-1 and/or Tim-3 positive as compared to CD8⁺ TIL (**Fig. 6**).
2. We measured PD-L1 in EMT6 cells by flow cytometry and found both constitutive and IFN-gamma inducible expression (**Fig. 6b**).
3. We measured PD-L1 in single cell suspensions from established EMT6 tumours and found high-level PD-L1 expression in F4/80⁺ macrophages and subsets expressing CD11b⁺MHCII⁺Ly6C⁺ (**Fig. 6c**). Lower level PD-L1 was also detected within the CD45 negative fraction (i.e., EMT6 tumour cells and CD45⁻ stromal cells; **Fig. 6c**).
4. Because many TIL were PD-1 positive, cultured EMT6 cells were PD-L1 positive and responded to IFN-gamma with high level PD-L1 expression, multiple tumour cell subsets were PD-L1 positive, and the PD-1/PD-L1 signaling axis within EMT6 tumours remained at least partly intact after LCL161±VSV treatment (**Fig. 6d,f**), we also tested whether αPD-1 therapy would alter outcomes to LCL161+VSV combination therapy. Indeed, we found αPD-1+LCL161+VSV triple combination therapy generated long-term, durable responses in nearly 90% of EMT6 tumour bearing mice (**Fig. 6a**). This is consistent with two very recent reports^{3,4} showing therapeutic synergy between SMC and αPD-1 in mouse models of multiple myeloma and glioma, respectively, and indicates to us that while LCL161+VSV can reverse a state of TIL exhaustion within EMT6 tumours, it is not complete and not entirely redundant with αPD-1 therapy.

4. Figure 2: The role of LCL161 on depleting macrophages is postulated from a series of in vitro experiments. Have the authors attempted depletion of these TAM to show that SMC therapy mimics the role of immune suppressive TAM?

Authors' response: In our original submission, we showed that LCL161 treatment causes the death of BMDMs grown in culture (**original Fig. S11**) and leads to partial TAM depletion within EMT6 tumours grown in mice (**original Fig. 2o**). The impetus for doing these experiments was twofold: *i.*) prior reports showing that IAP antagonism induces necroptosis of cultured macrophages⁵, and *ii.*) the changes we observed in EMT6 tumours post-LCL161 treatment that we hypothesized could reflect altered TAM functionality (**original Figs. 2a, b, f-i**). The latter point also prompted us to test whether LCL161 treatment alters macrophage function, and indeed we reported a direct effect of LCL161 treatment on the activation status of BMDMs grown in culture (**original Figs. 2j-m**). Based on both sets of results, we proposed that the immunosuppressive TME engendered by LCL161 treatment may be due to dual effects on TAM: polarizing them toward and “M1-like” activation state, and causing some of them to die.

Reviewer #1 has asked us to determine more directly whether the death of TAM is involved in the LCL161 effect *in vivo*, by testing whether their depletion phenocopies LCL161 treatment. The most

commonly used method to deplete TAM *in vivo* is by clodronate liposome (CL) treatment. We therefore addressed this comment by pre-treating mice with CL injected directly into the tumour prior to treatment with either VSV or LCL161. As expected, CL treatment caused a decrease in macrophages within the TME (**Fig. S16**), to a similar extent as LCL161 (**Fig. S15**). To our surprise, however, CL treatment did not improve VSV therapy (if anything it made it worse), and CL blunted the therapeutic effect of LCL161 (**Fig. S16**).

One interpretation of these results is that the partial depletion of TAM by LCL161 does not play an important role in the LCL161 mechanism of action, in terms of promoting better anticancer CD8+ T cell responses *in vivo*. We have therefore changed our manuscript to reflect new thinking.

Because of this result, we decided to perform additional experiments to further characterize the ability of LCL161 to activate/polarize TAM *in vivo* and directly in cell culture assays using BMDMs. First, we treated EMT6 tumour bearing mice with LCL161 and measured TAM phenotype by flow cytometry. In these experiments, we found that LCL161 shifts the population of TAM within EMT6 tumours away from an “M2-like” phenotype (**Fig. 3e**). Next, we cultured BMDMs and asked whether LCL161 altered the expression level of canonical cell surface activation markers, using flow cytometry. In these experiments, we found that a dose-dependent increase in class I and II MHC, CD40L and CD80 on BMDMs treated with LCL161 (**Fig. S17**). Finally, we cultured BMDMs and asked whether LCL161 affected their polarization, alone or upon stimulation with the M1-polarizing agent IFN-gamma and the M2 polarizing agent IL-4. Using flow cytometry or qPCR to measure the expression of the canonical M1 marker iNOS and M2 marker Arginase-1, we found that LCL161 promoted conversion to M1 and blunted conversion to M2 (**Fig. 3j-k**). Taken together, these data suggest that LCL161 treatment leads to TAM polarization by directly acting upon the macrophage itself. When taken together with a recent report by Chesi and colleagues (2016) showing that LCL161 can cause TAM activation/polarization, which leads to their enhanced ability to phagocytose cancer cells, we propose here that TAM polarization – not their depletion – underlie the TME altering effects of LCL161 *in vivo*, and have adjusted our manuscript accordingly

5. *Figure 3: The role of VSV as a potent systemic immune adjuvant is not really novel. In this model, where systemic VSV has no therapy effect alone, does the virus reach the tumors? The authors look in the TDLN but do not present data of virus in the tumor itself. Could the virus be activating TIL or de-suppressing TAM in the tumor?*

Authors' response: Experiments performed to address this comment have found that EMT6 tumours support a limited VSV infection as measured by plaque assay (**Fig. S18**). However, compared to some mouse tumours [e.g., CT-26^{6,7}] the infection is weak, likely because EMT6 cells can respond to type I interferon generated by immune and stromal cells within the inflamed tumour microenvironment⁸. Data in our original submission led to our interpretation that the tumour infection promotes T cell recruitment to the tumour as we showed that the VSV therapy leads to canonical T cell chemokine (CXCL9, CXCL10) and CD8+ T cell accumulation within the TME (**original Fig. 3a-b**), although no significant bearing on TIL exhaustion was observed (**original Fig. 3c**). In response to this referee's comments, we also have measured the effect of VSV treatment on TAM polarization by flow cytometry. As shown in **Fig. 4d**, VSV treatment has no significant effect on TAM orientation *in vivo*. Taken together, we interpret this data to show that a VSV infection of EMT6 tumours promotes T cell recruitment via chemokine secretion, but does not alter TAM number, polarization or TIL exhaustion.

Summary of remarks by Reviewer #2: This reviewer is very supportive of our manuscript, stating that “*the experiments are well performed, the data is very convincing and almost complete.*”

Furthermore, the reviewer agrees with the potential significance of the work, stating that “*the outcomes of this manuscript are of extreme importance to the field...*”. The concerns the reviewer raised with our paper relate both to *i.)* insufficient mechanistic insight, in particular with regards to our observation of divergent roles of CD4 T cells in the EMT6 vs. M3-9-M model system and the role of APC infections in secondary lymphoid organs by VSV, and *ii.)* over/misinterpretation of cell culture data, in particular with regards to macrophage activation/polarization.

1. In the two mouse models used, EMT6 and M3-9-M, a CD4 T cell depletion was performed. While in EMT6 CD4 depletion improved survival of the mice, this was not the case in M3-9-M. The authors suggest that in the case of EMT6 this is due to the depletion of high numbers of regulatory T cells that infiltrate the tumor and suppress immunity. It would be nice if the authors could substantiate this by - at least - showing how the CD8, CD4 and especially Treg infiltrate of EMT6 and M3-9-M tumors differ under no-treatment circumstances.

Authors’ response: CD4⁺ T cells are highly plastic, having phenotypes ranging from effector to helper to regulatory. Not surprisingly, their role in anticancer immunity is extremely complex. They’ve been documented to have direct and indirect cytotoxic effects against cancer cells, to condition the TME for better CD8⁺ T cell recruitment and activation, and/or to generate immune tolerance by suppressing the activity of CD8⁺ T and other immune cells⁹. Consistent with these diverse functions, it has commonly been observed that CD4 depletion abrogates the efficacy of immunotherapy in some mouse tumour models, yet leads to spontaneous tumour regression in others (see for examples refs^{10,11}). Although the mechanisms responsible for these different outcomes are complex and not entirely known, an explanation often put forth is the dependence of some models on effector CD4⁺ T cell activation for anticancer immunity in contrast to the hyper-polarization of CD4⁺ T cells toward a regulatory “immunosuppressive” phenotype in others.

To generate some insight into why the EMT6 model responded to CD4 depletion with spontaneous tumour regression whereas the M3-9-M tumours required CD4⁺ T cells for tumour clearance by LCL161±VSV therapy, we performed additional immunophenotyping as suggested by Reviewer #2. We report the following new data from those analyses (**Fig. S8**):

1. Total CD8⁺ T cells: CD8⁺ TILs are substantially more abundant in M3-9-M vs. EMT6 tumours (~24% vs. 6% of total CD45⁺ leukocytes).
2. Total CD4⁺ T cells: The number of CD4⁺ TILs were not different between M3-9-M vs. EMT6 tumours (~ 8% in both models).
3. CD4⁺ T cell phenotype: The large majority of CD4⁺ T cells in M3-9-M tumours were CD4⁺CD25⁺FoxP3⁻ (activated CD4 cells) whereas most CD4⁺ T cells in EMT6 tumours were CD4⁺CD25⁻FoxP3⁻ (naïve CD4 T cells, a T cell subset recently shown to be a precursor for regulatory T cells in breast tumours¹²).
4. Ratio of regulatory CD4⁺ to effector CD4⁺ or CD8⁺ TIL: The ratio of CD4⁺CD25⁺FoxP3⁺ to either CD4⁺CD25⁺FoxP3⁻ or CD8⁺ TIL was substantially higher in EMT6 vs. M3-9-M tumours.

While defining CD4⁺ T cell polarization based solely on CD25 and FoxP3 expression is not perfect, our analyses of these commonly used markers is consistent with our original suggestion that the CD4⁺

TIL fraction is generally more immunosuppressive within EMT6 vs. M3-9-M tumours, in which an activated subset appears to predominate. We do not know why CD4⁺ T cell subsets differentially populate EMT6 vs. M3-9-M tumours, but our data are consistent with the observation that CD4 depletion leads to spontaneous regression in EMT6 tumours (populated by immunosuppressive CD4⁺ T cells), but abrogates the effectiveness of the LCL161±VSV therapy in the M3-9-M tumours (populated by activated CD4⁺ T cells).

2. Lines 90-91, *Alternatively, VSV primes for better response to LCL161 (see also point 3)*

Authors' response: This is a good suggestion, and we have adjusted the manuscript to read "...whilst VSV serves to enhance or prime for a better response to LCL161."

3. Line 120, figure 2k: *while it is stated that LCL161 treated BMDC were hyperresponsive to VSV this cannot be concluded from the figure. There is no LCL161 only control shown, there is only a VSV-only control shown. Based on this one can only conclude that the increase response in cytokines is due to LCL161-induced hyper-responses after VSV treatment. This should be corrected (see also point 2).*

Authors' response: The data from Fig. 2j and 2k in our original submission were from the same experiment. We had separated them to allow us to use different scales for the y-axis, to highlight the observation that LCL161 alone leads to the secretion of proinflammatory cytokines. In response to this critique by Reviewer #2, we have now merged these two graphs (**Fig. 3f**) and also altered our description of the results, which now reads "*LCL161 treatment...leads to the secretion of proinflammatory cytokines, and their enhanced secretion upon VSV-inoculation*".

4. Line 123, *the authors claim that LCL161-treated BMDC displayed enhanced ability to present antigen (figure 2m) but this cannot be concluded. They only show that the treated BMDC have a stronger capacity to activate CD4 T cells. This may be due to higher expression of antigen (Signal 1) but it might also be that co-stimulation (signal 2) or cytokine production (signal 3) is altered, leading to better T cell activation. For the latter, they have the evidence.*

AND

5. *It would have been nice if the authors would have used in figure 2m CD8 T cells, since they show that their model is dependent on CD8 T-cell reactivity rather than CD4 T cells.*

Authors' response: These are good points. To address them, we have performed an experiment in which BMDMs were cultured with or without LCL161, pulsed with full-length OVA, and co-cultured with CD8⁺ T cells purified from transgenic OT-1 mice. This experiment showed that LCL161 treatment enhanced the activation of CD8⁺ T cells by BMDMs as measured by CD69 surface marker expression (**Fig. 3h**).

While speculative, we suspect that the enhanced antigen presentation by BMDM is a consequence of higher MHC levels, better co-stimulation and increased cytokine secretion, as we now have evidence that each of these are enhanced by LCL161 treatment *in vitro* (**Fig. 3f, S17**). However, our reason for doing these experiments was to test in a general sense whether LCL161 could directly alter the activation/polarization of macrophages, which all measures universally point towards. Regardless, we

have adjusted the written description of the antigen presentation data to now reads “*LCL161 treatment...heightens activation of T cells in class I and II MHC-restricted antigen presentation assays*”

6. Line 125, figure S11: the authors claim that the viability of LCL161 treated BMDC is decreased. First, this should be shown in a quantitative manner, one cannot really judge this by a picture. Secondly, how would this conceptually work, why activate and then kill M1 macrophages?

Authors’ response: We have quantified the effect of LCL161 on BMDM viability, which confirmed our visual observations (**Fig. S14**).

One possible explanation for the finding that SMC treatment causes activation and demise of some TAM is that the SMC-mediated activation of BMDMs lead to the secretion of factors (e.g., TNF) that can subsequently cause their death – because they are IAP-depleted. However, we have not pursued this (or other) possibilities, mostly because our intratumoural CL experiments (requested by Reviewer #1) showed that experimental TAM depletion does not phenocopy SMC therapy in terms of causing tumour regression. As explained in detail above and throughout the revised manuscript, we have therefore adjusted our interpretation of the role of TAM depletion by SMC to reflect these new data.

7. Lines 125-126: the authors state that sustained LCL161 treatment in vitro results in decreased cell number. Please explain “sustained”, and how different is this from their in vivo experiments, in other words how relevant is this experiment?

Authors’ response: We had used the word “sustained” to reflect the fact that we had treated the animals for 72 hours *in vivo* prior to analyzing tumours for TAM. We have removed the word to avoid confusion, and as discussed above have altered our interpretation of the role of TAM depletion by SMC treatment in the anticancer activity of SMC.

8. Line 150: without altering their overall numbers, should be changed into without significantly altering

Authors’ response: We have made this change to the manuscript, thank you.

9. Lines 150-159, while the authors make a strong case for the effect of VSV to infect antigen presenting cells thereby stimulating the production of large quantities of cytokines suggesting this may work as adjuvant for T cells, it was not proven beyond doubt. Potentially, final proof could be obtained by depletion of antigen presenting cells before treatment and then measure cytokines. For instance, through clodronate treatment.

Authors’ response: As suggested by reviewer #1, we performed an experiment in which CL was delivered *i.v.*, to deplete phagocytes in secondary lymphoid organs prior to treating with VSV^{AM51} (*i.v.*, 1e8 PFU). Blood was collected 6 h later for multiplex cytokine analyses. While CL led to the depletion (although not complete) of various APC subsets within the spleen and lymph node, it did not cause a significant reduction in the amounts of cytokines secreted into blood (see attached data). We speculate this occurs because either APC depletion by CL is insufficient, or in their partial absence other cells

can perform this important antiviral function in the mouse. Indeed, by intravital microscopy we have noticed that fluorescently-labelled VSV circulates in blood much longer in mice treated with CL (not shown), which may provide an opportunity for “abnormal” interactions between virions and different mouse cells that can also secrete cytokines upon a VSV encounter.

In the absence of CL treatment, the large majority of VSV infection occurs within the tumour (if sensitive), spleen and lymph nodes, with large amounts of virus being cleared by the liver and lung⁷. Within the spleen and lymph node, documented infections of VSV variants occur in CD169+ macrophages, DCs, and B cells¹³⁻¹⁶, APC that secrete large quantities of cytokines upon virus encounter. As pointed out by Reviewer #1, its already known that the VSV backbone can serve as a vaccine adjuvant. We believe that these facts, coupled with our data showing that

1. VSV^{ΔM51} infects APCs in spleen and LN (**Fig. 4f, 5d**)
2. VSV^{ΔM51} induces robust cytokine secretion into blood (**Fig. 4g**)
3. VSV^{ΔM51} non-specifically promotes anti-EMT6 T cell activation within the TdLN (**Fig. 4e**)
4. Systemic attenuation of VSV^{ΔM51} infection and cytokine secretion by LCL161 pre-treatment abrogates synergy between VSV^{ΔM51} and LCL161 (**Fig. 5**)

strongly supports our assertion that at least part of the VSV^{ΔM51} effect within this combination therapy is to act a systemic immune adjuvant for anticancer T cell responses.

1. Ravindranathan, S., Smith, S. G., Nguyen, K. & Zaharoff, D. A. Colony stimulating factors secreted by irradiated autologous tumour cell vaccines inhibits immunity. *Journal for ImmunoTherapy of Cancer* **3**, 448
2. Beug, S. T. *et al.* Smac mimetics and innate immune stimuli synergize to promote tumor death. *Nat. Biotechnol.* **32**, 182–190 (2014).
3. Chesi, M. *et al.* IAP antagonists induce anti-tumor immunity in multiple myeloma. *Nat. Med.* **22**, 1411–1420 (2016).
4. Beug, S. T. *et al.* Smac mimetics synergize with immune checkpoint inhibitors to promote tumour immunity against glioblastoma. *Nat Commun* **8**, (2017).
5. McComb, S. *et al.* cIAP1 and cIAP2 limit macrophage necroptosis by inhibiting Rip1 and Rip3 activation. *Cell Death Differ.* **19**, 1791–1801 (2012).
6. Naumenko, V, CJ, J. & DJ, M. in *The Tumour Microenvironment* **356**, 217–230
7. Breitbach, C. J. *et al.* Targeted inflammation during oncolytic virus therapy severely compromises tumor blood flow. *Mol. Ther.* **15**, 1686–1693 (2007).
8. Liu, Y.-P., Suksanpaisan, L., Steele, M. B., Russell, S. J. & Peng, K.-W. Induction of antiviral genes by the tumor microenvironment confers resistance to virotherapy. *Sci Rep* **3**, 2375 (2013).
9. Kim, H.-J. & Cantor, H. CD4 T-cell subsets and tumor immunity: the helpful and the not-so-helpful. *Cancer Immunol Res* **2**, 91–98 (2014).
10. Lawson, K. A. *et al.* Repurposing Sunitinib with Oncolytic Reovirus as a Novel Immunotherapeutic Strategy for Renal Cell Carcinoma. *Clin. Cancer Res.* **22**, 5839–5850 (2016).
11. Sharma, R. K., Yolcu, E. S., Srivastava, A. K. & Shirwan, H. CD4+ T cells play a critical role in the generation of primary and memory antitumor immune responses elicited by SA-4-1BBL and TAA-based vaccines in mouse tumor models. *PLoS ONE* **8**, e73145 (2013).
12. Su, S. *et al.* Blocking the recruitment of naive CD4(+) T cells reverses immunosuppression in breast cancer. *Cell Res.* **2**, e25444 (2017).
13. Junt, T. *et al.* Subcapsular sinus macrophages in lymph nodes clear lymph-borne viruses and present them to antiviral B cells. *Nature* **450**, 110–114 (2007).
14. Iannaccone, M. *et al.* Subcapsular sinus macrophages prevent CNS invasion on peripheral infection with a

- neurotropic virus. *Nature* **465**, 1079–1083 (2010).
15. Honke, N. *et al.* Enforced viral replication activates adaptive immunity and is essential for the control of a cytopathic virus. *Nat. Immunol.* **13**, 51–57 (2012).
 16. Bridle, B. W. *et al.* Privileged Antigen Presentation in Splenic B Cell Follicles Maximizes T Cell Responses in Prime-Boost Vaccination. *J Immunol* **196**, 4587–4595 (2016).

Effect of i.v. CL treatment in the spleen

- No Tx
- Control liposome
- ▲ Clodronate liposome

Effect of i.v. CL treatment on VSV-induced cytokine secretion into blood

REVIEWERS' COMMENTS:

Reviewer #1 (Remarks to the Author):

The authors have responded extensively to my comments with extra data and extra discussion.

Reviewer #2 (Remarks to the Author):

The authors have tremendously improved their manuscript and took care to address all remarks and I have only a few small left.

1. Figure S8 shows the immunophenotyping of the two tumors. I agree with the interpretations except of that with respect to the Foxp3+ cells. The flow cytometry plots showing FOxp3+ staining are not convincingly positive for FOXP3. If this is the staining they had, I would advice to remove Figure S8b and the flowcytometry, as well as the text on this. The remainder still addresses the earlier question.
2. Figure S14. The LDH assay is a way to quantify cell death. I interpret the data shown as "hardly any killing". Hence, I would not use this for the statement at line 130, rather I would use it to back up the statement that LCL161 does not deplete the TAM by killing.
3. Lines 141-143. The authors state that some TAM are required for tumor regression. As a note to the authors, the notion that TAMs are required for tumor regression was also made by another group, see van der Sluis et al Cancer Immunol Res 2015.

Reviewers' comments and response by authors:

Reviewer #2 (Remarks to the Author):

The authors have tremendously improved their manuscript and took care to address all remarks and I have only a few small left.

1. Figure S8 shows the immunophenotyping of the two tumors. I agree with the interpretations except of that with respect to the Foxp3+ cells. The flow cytometry plots showing FOxp3+ staining is not convincingly positive for FOXP3. If this is the staining they had, I would advise to remove Figure S8b and the flow cytometry, as well as the text on this. The remainder still addresses the earlier question.

Authors' response: This is a fair point, and we have removed Figure S8b and the flow cytometry plots as suggested. We have adjusted the text accordingly (lines 98-105)

2. Figure S14. The LDH assay is a way to quantify cell death. I interpret the data shown as "hardly any killing". Hence, I would not use this for the statement at line 130, rather I would use it to back up the statement that LCL161 does not deplete the TAM by killing.

Authors' response: Another fair point, and we have adjusted the text to reflect this interpretation (lines 150-51).

3. Lines 141-143. The authors state that some TAM are required for tumor regression. As a note to the authors, the notion that TAMs are required for tumor regression was also made by another group, see van der Sluis et al Cancer Immunol Res 2015.

Authors' response: We thank the reviewer for pointing us toward this reference, and have added it to our manuscript (line 159).